# Dimension-Independent Rates for Structured Neural Density Estimation

**Robert A. Vandermeulen**    **Wai Ming Tai**    **Bryon Aragam**

## Abstract

We show that deep neural networks can achieve dimension-independent rates of convergence for learning structured densities typical of image, audio, video, and text data. For example, in images, where each pixel becomes independent of the rest of the image when conditioned on pixels at most $t$ steps away, a simple $L^2$-minimizing neural network can attain a rate of $n^{-1/((t+1)^2+4)}$, where $t$ is independent of the ambient dimension $d$, i.e. the total number of pixels. We further provide empirical evidence that, in real-world applications, $t$ is often a small constant, thus effectively circumventing the curse of dimensionality. Moreover, for sequential data (e.g., audio or text) exhibiting a similar local dependence structure, our analysis shows a rate of $n^{-1/(t+5)}$, offering further evidence of dimension independence in practical scenarios.

## 1. Introduction

Deep learning has emerged as a remarkably effective approach to numerous statistical problems that were historically extremely challenging, especially in high-dimensional settings. Deep generative models, for example, can approximate densities with thousands or millions of dimensions using merely a few million data points (Oussidi & Elhassouny, 2018; Ho et al., 2020; Cao et al., 2024). This stands in stark contrast to standard density estimation theory, which would demand astronomical sample sizes due to the curse of dimensionality. At the same time, it is known that if a density satisfies certain assumptions such as monotonicity, convexity, sparsity, mixtures, or additivity, the curse of dimensionality can be mitigated. Perhaps the most widely accepted explanation for deep learning's ability to circumvent this curse is the *manifold hypothesis* (Bengio et al., 2013; Brahma et al., 2016). This hypothesis posits that, despite a distribution's ambient space being high-dimensional,

the mass of the density is heavily concentrated around a lower-dimensional subset of that space, such as an embedded manifold. As we will show later, for complex data types of significant interest—images, video, sound, and text—this assumption is intimately linked to spatio-temporal locality. For instance, in data such as images, video, sound, and text, spatio-temporal locality implies that nearby covariates—e.g. pixel intensities at adjacent positions in an image—tend to be strongly dependent, suggesting that the data is effectively constrained to a lower-dimensional subspace.

This paper investigates the benefits of leveraging the converse structure: The *independence* of spatio-temporally distant covariates. Covariates that are spatio-temporally distant often exhibit near-independence, particularly after conditioning on nearby covariates. Consider a sound recording: two one-second segments separated by a minute might share common elements, such as the same speaker. However, given the intervening minute of audio, these segments become effectively independent. The minute-long interval contains sufficient information to render the separated segments mutually uninformative. This principle of conditional independence extends to various data types, including images, where pixels far apart tend to be independent when conditioned on the surrounding region.

For spatial data—focusing on images for concreteness—it is well known that a pixel becomes independent of the rest of the image when conditioned on all pixels within a certain distance $t$, following an argument similar to the one made for sound segments above. We show that, under this assumption, density estimation using a neural network with a simple $L^2$-minimizing loss achieves a rate of approximately $n^{-1/((t+1)^2+4)}$. In contrast the standard rate in nonparametric density estimation for the same class of densities is $n^{-1/(2+d)}$, where $d$ represents the total number of covariates. This implies that the effective dimension is of order $t^2$ instead of $d$. For comparison, in images, $t$ is typically on the order of 3-8 (see Section 3), whereas $d$ is the full image dimensionality given by width times height, which would be substantially larger than $t$ in practice. For example, consider simple benchmark datasets like MNIST ($28 \times 28$, $d = 784$) or CelebA ($256 \times 256$, $d = 65\,536$). Similarly, for sequential data, a dimension-independent rate of $n^{-1/(t+5)}$ is achieved.

.    Correspondence to:    Robert A. Vandermeulen <robert.anton.vandermeulen@gmail.com>.

*Proceedings of the 42nd International Conference on Machine Learning*, Vancouver, Canada. PMLR 267, 2025. Copyright 2025 by the author(s).

A key aspect of the results presented here is that $t$ remains largely independent of the overall size of the image or the total duration of the temporal data. Returning to the sound recording example, a one-second segment may become independent of the rest of the recording when conditioned on the minute of audio preceding and following it. Crucially, this conditional independence holds regardless of whether the full recording spans three minutes or an hour.

## 2. Background and Related Work

In this section, we lay the foundation for our main results by introducing key concepts and related work. We begin by discussing traditional approaches to nonparametric density estimation and their limitations, particularly the curse of dimensionality. We then explore the manifold hypothesis, a widely accepted explanation for the success of deep learning in high-dimensional settings, along with other structural hypotheses used in nonparametric density estimation. Following this, we review the basics of Markov random fields (MRFs) and their applications in modeling various types of data, including images and sequential information. This background will provide the necessary context for understanding the novelty of our approach, which leverages MRFs to achieve dimension-independent convergence rates in density estimation. While our framework offers an alternative perspective to the manifold hypothesis, it neither precludes nor implies it; rather, the two assumptions address different structural properties of high-dimensional data.

### 2.1. Nonparametric Density Estimation

Density estimation is the task of estimating a $d$-dimensional target probability density $p$ from observed data, $\mathbf{x}_1, \ldots, \mathbf{x}_n \overset{\text{iid}}{\sim} p$. Of course, this is a classical problem for which we do not intend to provide a comprehensive overview, and instead refer readers to books such as (Devroye & Gyorfi, 1985; Devroye & Lugosi, 2001; Tsybakov, 2009) for additional background. For general densities $p$, nonparametric density estimators like kernel density estimators or histograms (Devroye & Gyorfi, 1985; Devroye & Lugosi, 2001) converge to $p$ for *any* density given sufficient data, but notably suffer from the curse of dimensionality. For instance, when $p$ is Lipschitz continuous[1], the $L^1$ error $\int |p(x) - \hat{p}_n(x)| \, dx = \|p - \hat{p}_n\|_1$, converges at the rate $O(n^{-1/(2+d)})$. This *nonparametric rate* is known to be optimal for Lipschitz continuous densities, and numerous studies over the past decade have established that neural networks and generative models can achieve this optimal rate (Liang, 2017; Singh et al., 2018; Uppal et al., 2019;

Kuzborskij & Szepesvári, 2022; Oko et al., 2023; Zhang et al., 2024; Kwon & Chae, 2024). This rate implies that the sample complexity grows *exponentially* in the dimension $d$, making the success of deep neural networks for estimating densities with millions of dimensions all the more remarkable. Various types of structure have been studied to improve the rate of convergence. For example, Hall & Zhou (2003); Hall et al. (2005); Vandermeulen & Ledent (2021); Vandermeulen (2023); Chhor et al. (2024) show that one can achieve dimension-independent rates of convergence for nonparametric density estimation by assuming a multiview model—a type of low-rank structure. However, in the context of deep learning, this phenomenon is most often explained via the *manifold hypothesis*.

### 2.2. Manifold Hypothesis

The manifold hypothesis posits that many high-dimensional real-world distributions concentrate around lower-dimensional spaces, such as submanifolds of the ambient space. The success of deep learning methods in handling high-dimensional data, such as images, videos, and audio, is frequently attributed to the manifold hypothesis.

For example, Pope et al. (2021) determined that the intrinsic dimension of the ImageNet dataset lies between 25 and 40 dimensions, significantly lower than its ambient dimension. Further supporting this hypothesis, Carlsson et al. (2008) discovered that the set of $3 \times 3$ pixel patches from natural images concentrates around a 2-dimensional manifold. Theoretically, distributions concentrating around lower-dimensional subsets of the ambient space have been shown to yield improved estimation properties. For instance, Weed & Bach (2019) demonstrated that while the empirical distribution typically converges at rate $n^{-1/d}$ to the true distribution in Wasserstein distance, when the data has a lower $d'$-dimensional structure, it converges at the faster rate of $n^{-1/d'}$. Similar results illustrating the manifold hypothesis and its benefits abound in the literature (Pelletier, 2005; Ozakin & Gray, 2009; Jiang, 2017; Schmidt-Hieber, 2019; Nakada & Imaizumi, 2020; Berenfeld et al., 2022; Chae et al., 2023; Jiao et al., 2023; Tang & Yang, 2024). Another line of related work uses Barron functions, which are a class of functions inspired by neural networks, to achieve dimension-free rates (Barron, 1993; Klusowski & Barron, 2018; Ma et al., 2022; Cole & Lu, 2024), in contrast to our use of Lipschitz-type assumptions.

While the manifold hypothesis explains local dependencies, it's worth considering scenarios that deviate from this model. For example, the manifold hypothesis *cannot* be satisfied when covariates are independent. This is easily seen between two independent covariates $x \sim p_x$ and $y \sim p_y$, whose joint density $p_{x,y}(x, y) = p_x(x) p_y(y)$ fills a rectangle in their product space. More generally, if a probability

---

[1] A function $f : \mathbb{R}^d \to \mathbb{R}$ is *Lipschitz continuous* if there exists $L \geq 0$ such that $|f(x) - f(y)| \leq L \|x - y\|_2$ for all $x, y$. Our results extend to more general families (e.g. Hölder), however, we present our main results in the Lipschitz setting for simplicity.

measure is supported on a manifold, there is a limit on how many covariates can be mutually independent. We formalize this with the following proposition.

**Proposition 2.1.** *Let $M \subset \mathbb{R}^d$ be a regular submanifold of dimension $d' < d$. Let $X = (X_1, \ldots, X_d)$ be a random vector distributed according to a probability measure $\mu$ supported on $M$, such that each marginal $X_i$ has a non-degenerate probability density function $p_i$. Then, for any subset of $d' + 1$ coordinates $I_1, \ldots, I_{d'+1}$, the random variables $X_{I_1}, \ldots, X_{I_{d'+1}}$ cannot be mutually independent.*

The proof of this proposition can be found in the appendices. This result formalizes the intuition that in higher dimensions, the manifold hypothesis limits the amount of independence that is allowed. In this sense, the manifold hypothesis is complementary to our MRF approach.

Finally, while the manifold hypothesis is arguably the most widely accepted explanation for the learnability of high-dimensional models, particularly within the deep learning community, it is far from the only structural assumption considered in the broader nonparametric estimation literature. Various alternative approaches have been explored such as monotonicity (Groeneboom, 1984), convexity (Groeneboom et al., 2001), log-concave (Samworth, 2018), sparsity (Liu et al., 2007), mixtures (Genovese & Wasserman, 2000), and additive models (Stone, 1985). Crucially, with the exception of additivity and sparsity, these assumptions do not address the curse of dimensionality. While sparsity is a special case of the manifold hypothesis (e.g. the manifold is a linear subspace or union of linear subspaces), realistic models of image, audio, video, and text data do not satisfy additivity assumptions. From this perspective, we propose a novel approach to mitigating the curse of dimensionality that is on the one hand much more general than additivity while at the same time complementing the manifold hypothesis.

### 2.3. Markov Random Fields

Our intuition about natural images suggests that pixels that are distant from one another tend to become more independent. This observation naturally leads to modeling the space of images as ab MRF, where local dependencies are captured while allowing for independence between distant pixels (Li, 1994; 2009; Blake et al., 2011). Here, we briefly review the requisite concepts from MRFs as they are used in vision and imaging applications.

A *Markov random field* (MRF) consists of a random vector $\mathbf{x} = (x_1, \ldots, x_d)$ and a graph $\mathcal{G} = (V, E)$, where the graph's vertices correspond to the entries of the random vector, i.e., $V = \{x_1, \ldots, x_d\}$. The graph encodes information about the conditional independence of the vector's entries. For a set $A = \{a_1, \ldots, a_{d'}\} \subset \{1, \ldots, d\}$, let $\mathbf{x}_A = (x_{a_1}, \ldots, x_{a_{d'}})$. Given three disjoint subsets $A, B, C$

of $\{x_1, \ldots, x_d\}$, the graph $\mathcal{G}$ indicates that the random vectors $\mathbf{x}_A$ and $\mathbf{x}_B$ are conditionally independent given $\mathbf{x}_C$ if there is no path from $A$ to $B$ that doesn't pass through $C$.

Consider a simple example with random variables x, y, and z, where $x = y + \epsilon_x$ and $z = y + \epsilon_z$, with $\epsilon_x$, $\epsilon_z$, and y being jointly independent. In this scenario, the distributions of x and z are conditionally independent given y. The corresponding MRF for this example is given by:

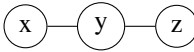

While an MRF conveys information about *conditional independence*, the absence of such information in the MRF does not necessarily imply dependence in the actual data. In other words, covariates can be conditionally independent in reality even if this independence is not explicitly represented in the MRF. Thus, the MRF provides a conservative model of independence relationships, capturing known or assumed conditional independencies without ruling out additional independencies that may exist in the data. Consequently, any random vector associated with a complete graph—where all vertices are adjacent to one another—is a valid MRF, since it provides no information about the independence of the covariates. This is because every vertex is connected to every other vertex, so removing any number of vertices will never separate the graph into multiple components. Because it conveys no information about the conditional independence of the covariates, it even applies to a random vector where all entries are independent.

*Remark* 2.2. After the initial posting of our paper, we were made aware of the related work by Bos & Schmidt-Hieber (2023) which proposes a supervised approach to density estimation that also leverages the Markov assumption to break the curse of dimensionality in density estimation.

### 2.4. MRFs in Applications

MRFs are widely used in applications; Figure 1 illustrates three of the most common families of MRFs.

**Paths (a.k.a. Markov chains)** One of the most well-known MRFs is the Markov chain. A Markov chain of length $d$ corresponds to the "path" graph $L_d$ of $d$ random variables. The above example with x, y, z corresponds to the graph $L_3$ and the MRF corresponding to $L_4$ shown in Figure 1. In a Markov chain, the indices are often interpreted as a time parameter. A classic example is a gambling scenario: A person's money at time $t + 1$ is conditionally independent of their total value at time $t - s$ (for $s > 0$), given their value at time $t$. This property, known as the Markov property, encapsulates the idea that the future state depends only on the present state, not on past states. Markov chains have a long history of use for modeling sequential information,

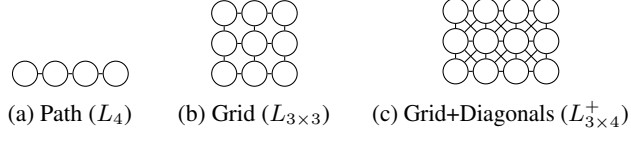

(a) Path ($L_4$)  (b) Grid ($L_{3\times3}$)  (c) Grid+Diagonals ($L_{3\times4}^+$)

Figure 1: Examples of common Markov random field graphs.

including audio and text data.

**Grids (a.k.a. lattices)**  Beyond sequential data, MRFs have seen significant use in image processing. In this work, we focus on grayscale images for simplicity, bearing in mind that the results extend to RGB/color images as well. For image processing, the classic MRF model consists of a random variable that is a 2-dimensional grid $\mathbf{X} = [\mathrm{X}_{i,j}]_{i,j}$ and a graph $\mathcal{G}$ where all pixels adjacent in $\mathbf{X}$ are also adjacent in $\mathcal{G}$. Figure 1 contains one example from two different types of grid graphs: one standard "grid" graph $L_{3\times3}$ and one "grid with diagonals" graph $L_{4\times3}^+$, where the subscripts indicate the number of rows and columns of vertices, respectively. For the remainder of this work, our references to "grid" graphs encompass both variants—those with and without diagonal connections—unless explicitly stated otherwise.

Such models have seen wide use in image processing and computer vision (see Li, 1994; 2009; Blake et al., 2011, for an overview). Denoising stands as perhaps the most common application of MRFs in image processing. This approach assumes that each pixel is best predicted using just its neighbours and ignoring the rest of the image. While this model proves effective for mitigating phenomena like additive white noise (Keener, 2010), it falls short as a comprehensive image model. Similarly, the path graph, often used for sequential data, oversimplifies the complex dependencies in realistic sequential data.

## 3. Improving The Path and Grid Markov Random Field Models

While standard path and grid MRF models may suffice for correcting extremely local or high-frequency noise in sequential or spatial data, they fall far short of capturing the true distribution of complex data types. Consider, for example, audio data consisting of 21-second clips where the middle second is missing and needs to be predicted. According to the path MRF model, this missing second would depend solely on the audio samples directly preceding and following it. Consequently, under a Markov chain (i.e. path MRF) model, the remaining 20 seconds of audio (less two samples) would be deemed completely uninformative for predicting the middle second, given these two adjacent samples. In this section we propose a richer model for capturing longer-range dependencies via power graphs.

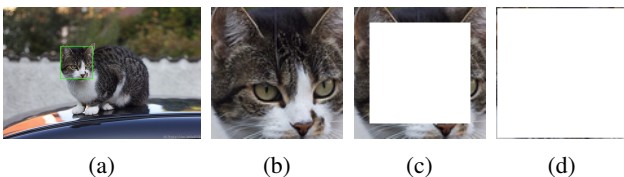

(a)  (b)  (c)  (d)

Figure 2: The leftmost image (a) is a $640 \times 427$ pixel photograph from the COCO 2014 dataset (Lin et al., 2014). Image (b) shows an enlarged version of the $102 \times 102$ pixel region outlined in (a). Images (c) and (d) display the 12-pixel and 1-pixel width borders of that region, respectively. Modeling this image with an MRF graph $L_{640\times427}$ or $L_{640\times427}^+$ would imply that the distribution of the missing interior in (d) depends exclusively on its 1-pixel wide border, with the rest of the image in (a) being uninformative for predicting this interior region. In contrast, predicting the interior using the 12-pixel border in (c) is more reasonable. This scenario corresponds to models like $L_{640\times427}^6$ or $\left(L_{640\times427}^+\right)^6$, which capture more extensive local dependencies. It's important to note that for the MRF model to hold, the interior doesn't need to be *deterministically* constructed from the surrounding pixels. Rather, the surrounding pixels need only provide sufficient information about the interior (e.g., that it's a cat's face) such that the rest of the image doesn't contribute any additional information for predicting the interior region.

### 3.1. Long-Range Dependencies

Simplistic models such as path and grid MRFs fail to capture the richer, longer-range dependencies present in real-world data. For example, in the audio example above, the content of the missing second is likely influenced by a broader context than just its immediate neighbours. For instance, the rhythm or theme established in the preceding few seconds, or the anticipation of what follows immediately after, could be crucial for predicting the missing segment. This moderately broader context is entirely discarded by the basic path MRF. Similarly, for image data, the standard grid MRF model suggests that a region of an image depends only on its immediate bordering pixels. However, realistic images often exhibit patterns and structures that span multiple pixels in various directions. For example, the edge of an object or a gradient in lighting might extend across several pixels, creating dependencies that the basic grid model fails to capture. Figure 2 illustrates this concept concretely, demonstrating the effects of different MRF models on image inpainting tasks and highlighting the implications of varying levels of contextual information. These limitations motivate the need for more sophisticated MRF models where segments or regions are more extensively connected, allowing for the incorporation of relevant contextual information without necessarily spanning the entire dataset.

To model sequential and spatial data more realistically, we

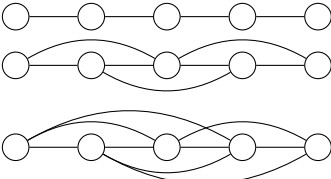

Figure 3: Illustrations of a path graph and its powers. Top: The path graph $L_5$. Middle: The power graph $L_5^2$. Bottom: The power graph $L_5^3$. In $L_5$, only immediately contiguous vertices are connected. In $L_5^2$, every group of three contiguous vertices forms a complete subgraph. In $L_5^3$, every group of four contiguous vertices forms a complete subgraph. This progression demonstrates increasing connectivity among nearby vertices in the graph.

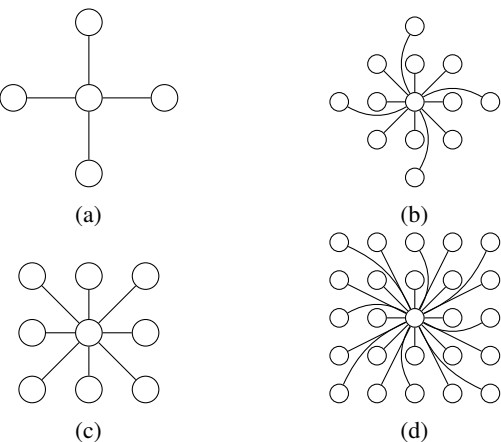

| (a) | (b) |
| (c) | (d) |

Figure 4: Comparison of vertex neighborhoods in different graph structures. (a) Neighborhood of a vertex in a standard grid graph $L_{d \times d}$. (b) Neighborhood of the same vertex in the power graph $L_{d \times d}^2$. (c) Neighborhood of a vertex in a grid graph with diagonals $L_{d \times d}^+$. (d) Neighborhood of the same vertex in the power graph $(L_{d \times d}^+)^2$.

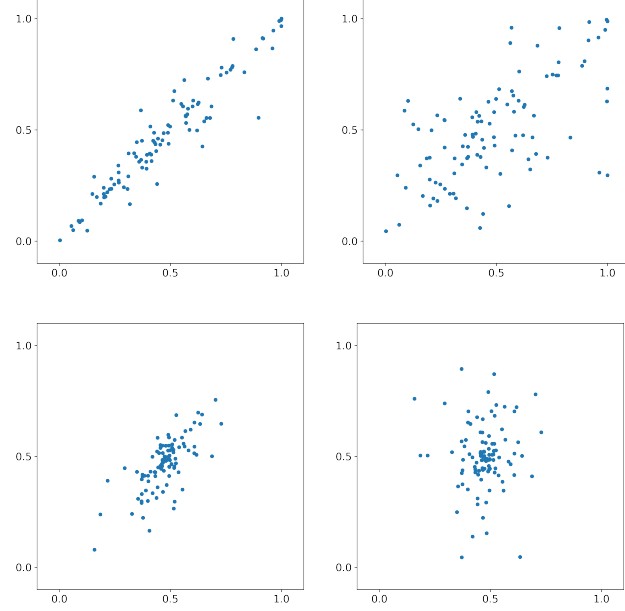

Figure 5: Top row: Scatterplots comparing the grayscale values of pixels in CIFAR-10. The **left** image shows (8,8) vs (8,9) and the **right** shows (8,8) vs (9,12).
Bottom row: These show the same scatterplots, conditioned on pixel (9,8) having a value approximately equal to 0.48 (the median value for this pixel across the dataset).
These plots demonstrate how pixel correlations decrease with distance and how conditioning on a neighboring pixel can significantly reduce correlations, supporting the use of Markov Random Field models for image data. Similar plots for the COCO dataset and for audio data from the Google Speech Commands Dataset (Warden, 2018) can be found in Appendix E.

propose using the "power graph" of the path and grid models. For a graph $\mathcal{G}$, the power graph $\mathcal{G}^t$ with $t \in \mathbb{N}$ is defined as the graph where we add an edge between every pair of vertices within $t$ steps of each other in $\mathcal{G}$, with $\mathcal{G}^1 = \mathcal{G}$. Figures 3 and 4 illustrate this concept using path graphs and grid graphs, respectively. This construction causes contiguous sections of sequences and patches of grids to become fully connected, as demonstrated in Figure 3.

Applying this power graph concept to a grid graph assumes that local patches of images are highly dependent, making no assumptions about conditional independence within a patch. It also implies that, in general, distant regions of an image become independent as the distance between them increases, and that these regions are independent when conditioned on a sufficiently wide separating region of pixels.

### 3.2. Experimental Validation

While MRFs are extremely well-established in image processing (Li, 2009; Blake et al., 2011), it is nonetheless instructive and informative to experimentally validate these assumptions using natural images. We provide an example using CIFAR-10 here, while similar experiments using the COCO and Google Speech Commands datasets are presented in Appendix E. The top row of Figure 5 shows the grayscale values of pixel (8,8) versus selected other pixels for 100 randomly chosen images from the CIFAR-10 training dataset. The bottom row repeats this experiment, but conditioned on the value of the adjacent pixel (9,8) being near its median value.

These experiments reveal that, when conditioned on the adjacent pixel, the dependence (as measured by correlation) decreases significantly. Notably, pixels (8,8) and (9,12) appear almost completely independent when conditioned on

pixel (9,8). This provides strong evidence for the validity of the MRF model. The MRF model predicts that (8,8) and (9,12) should be independent when conditioned on surrounding pixels (the number of which depends on the graph power of the MRF graph). Remarkably, we observe that pixels appear independent when conditioned on just a *single* adjacent pixel, suggesting that the grid MRF assumption may be even more conservative than necessary.

### 3.3. Comparison to the Manifold Hypothesis

The power graph extension of path and grid MRFs presents a fundamentally different perspective on modeling high-dimensional data compared to the widely accepted manifold hypothesis. While the manifold hypothesis posits that high-dimensional data concentrates around lower-dimensional structures, our MRF approach embraces the full dimensionality of the data, focusing instead on the independence structure between variables. This model aligns well with the observed structure in various data types, capturing local dependencies while allowing for long-range independencies. For sequential data such as audio or text, it accounts for strong dependencies between nearby elements while acknowledging the decreasing influence of distant context. In spatial data like images, it models high correlation between neighbouring pixels and gradual decorrelation as distance increases. Our experimental results provide compelling evidence for the MRF model's validity. The observed conditional independence between distant pixels, given intervening pixels, supports the fundamental assumptions of the power graph model.

Moreover, in special cases, the manifold hypothesis can be seen as closely linked to conditional independence in MRFs. In images, for example, adjacent pixels $x_i$ and $x_j$ typically have similar values, causing the dataset to concentrate towards the linear subspace $x_i = x_j$, which is a submanifold. The top left image in Figure 5 also illustrates this idea, where a strong concentration along the diagonal is evident. Nonetheless, these assumptions need not align in general; see Appendix F for illustrative examples of how these assumptions can vary independently.

Evidently, the manifold hypothesis cannot replace the MRF approach, and vice versa. Thus, it's important to note that our results are not meant to supersede the manifold hypothesis, but rather to augment and complement it. The manifold hypothesis explains sample efficiency from local structure, while the MRF model provides additional model efficiency from a global perspective. Together, they provide a more comprehensive framework for understanding high-dimensional data.

Remarkably, in the following section, we will demonstrate that under these MRF assumptions, there exist estimators based on neural networks with standard loss functions (e.g.

squared loss) that can achieve dimension-independent rates of convergence for density estimation. This result is particularly significant as it suggests a path to overcoming the curse of dimensionality in high-dimensional density estimation tasks. By focusing on independence structures rather than dimension reduction, our approach offers a novel explanation for the success of deep learning methods in processing complex, high-dimensional data, complementing and contrasting with the insights provided by the manifold hypothesis.

## 4. MRF-Based Density Estimation with Neural Networks

After introducing some theoretical prerequisites we present our main result, showing that a structured density $p$ can be estimated at a rate that depends only on $t$. Our estimator is based on a neural network trained to minimize an empirical estimate of the $L^2$ distance between the network's output and the target distribution $p$.

### 4.1. Structured Neural Density Estimation

Our estimator is based on the classical Hammersley-Clifford Theorem (Hammersley & Clifford, 1971). Before presenting the theorem we must review a few concepts. A graph $\mathcal{G}$ is called *complete* if every vertex is adjacent to every other vertex. For a graph $\mathcal{G} = (V, E)$ a *clique* is a complete subgraph, i.e., $\mathcal{G}' = (V', E')$ with $V' \subset V$ and $E' \subset E$, that is complete. A *maximal clique* of a graph is a set of cliques which are not contained within another clique. Observe that maximal cliques of the same graph can have different numbers of vertices. See Figure 6 for examples of maximal cliques. The collection of the maximal cliques of a graph will be denoted $\mathcal{C}(\mathcal{G})$.

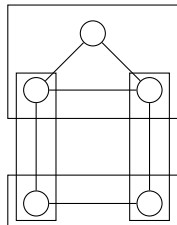

Figure 6: A graph with maximal cliques denoted by surrounding rectangles.

**Proposition 4.1** (Hammersley & Clifford, 1971). *Let $\mathcal{G} = (V, E)$ be a graph and $p$ be a probability density function satisfying the Markov property with respect to $\mathcal{G}$. Let $\mathcal{C}(\mathcal{G})$ be the set of maximal cliques in $\mathcal{G}$. Then*

$$p(\boldsymbol{x}) = \prod_{V' \in \mathcal{C}(\mathcal{G})} \psi_{V'}(\boldsymbol{x}_{V'}),$$

*where $\boldsymbol{x}_{V'}$ are the indices of $\boldsymbol{x}$ corresponding to $V'$.*

For neural networks we investigate estimators of the form:

$$\hat{p}(\boldsymbol{x}) = \prod_{V' \in \mathcal{C}(\mathcal{G})} \widehat{\psi}_{V'}(\boldsymbol{x}_{V'}),$$

where $\widehat{\psi}_{V'}$ are ReLU networks with architectures dependent only on $\mathcal{G}$ and the sample size $n$. The weights are constrained to $[-1, 1]$, effectively implementing weight decay via constrained optimization rather than norm penalization.

Our estimator minimizes the integrated squared error between $p$ and our estimator $\hat{p}$:

$$\int \left( p\left(\boldsymbol{x}\right) - \hat{p}\left(\boldsymbol{x}\right) \right)^2 d\boldsymbol{x}$$

$$= \int p(\boldsymbol{x})^2 d\boldsymbol{x} - 2 \int p(\boldsymbol{x})\hat{p}(\boldsymbol{x})d\boldsymbol{x} + \int \hat{p}(\boldsymbol{x})^2 d\boldsymbol{x}. \quad (1)$$

In empirical minimization, the first term of equation 1 is constant. The second term can be estimated using the law of large numbers:

$$\int p(\boldsymbol{x})\hat{p}(\boldsymbol{x})d\boldsymbol{x} = \mathbb{E}_{\mathbf{x} \sim p}\left[\hat{p}(\mathbf{x})\right] \approx \frac{1}{n} \sum_{i=1}^{n} \hat{p}(\mathbf{x}_i),$$

where $\mathbf{x}_1, \ldots, \mathbf{x}_n \overset{\text{i.i.d.}}{\sim} p$, i.e., the training data. Let $U_d$ be the $d$-dimensional uniform distribution on the unit cube and $\epsilon_1, \epsilon_2, \ldots, \epsilon_{n'} \overset{\text{i.i.d.}}{\sim} U_d$. Then:

$$\int \hat{p}(\boldsymbol{x})^2 d\boldsymbol{x} = \mathbb{E}_{\epsilon \sim U_d}\left[\hat{p}(\epsilon)^2\right] \approx \frac{1}{n'} \sum_{i=1}^{n'} \hat{p}(\epsilon_i)^2.$$

From these two estimates we see that estimating the $L^2$ distance between an estimator and a target density (ignoring the constant $\|p\|_2^2$ term) is tractable.

### 4.2. Main Result

We now present our theorem on the convergence rate for $L^2$-minimizing neural network-based density estimators:

**Theorem 4.2.** *Let $\mathcal{G} = (V, E)$ be a finite graph and $r$ be the size of the largest clique in $\mathcal{G}$. There exists a known sequence of architectures $\mathcal{F}^*$ such that for*

$$\hat{p}_n = \arg\min_{f \in \mathcal{F}^*} \left( \|f\|_2^2 - \frac{2}{n} \sum_{i=1}^{n} f(\mathbf{x}_i) \right),$$

*where $\mathbf{x}_1, \ldots, \mathbf{x}_n \overset{\text{i.i.d.}}{\sim} p$, we have*

$$\|p - \hat{p}_n\|_1 \in \widetilde{O}_p\left(n^{-1/(4+r)}\right),$$

*for any Lipschitz continuous, positive density $p$ satisfying the Markov property with respect to $\mathcal{G}$.*

The proof of the theorem, based on results from Schmidt-Hieber (2017), details the architectures and specifies how their parameters scale with the sample size. The minimax rate for density estimation on $d$-dimensional densities is $O\left(n^{-1/(2+d)}\right)$, so the "effective dimension" of an estimating a density using the estimator from the theorem above is $r + 2$. Consequently we see that the rate of convergence for density estimation can be greatly improved for MRFs with certain graphs $\mathcal{G}$. In the next section, we explore the implications of this result for the estimation of spatio-temporal data.

*Remark* 4.3. Our results are not specific to ReLU networks and can be extended to any class of neural networks for which appropriate approximation bounds are available. For example, our results can be extended to various non-ReLU networks by using the results of Ohn & Kim (2019).

### 4.3. Consequences of Main Result

Our results indicate that the effective dimension of density estimation problems in spatio-temporal data, under MRF assumptions, is determined by the size of the largest clique in the underlying graph. This has significant implications for modeling and learning in domains such as video, audio, and sequential data, where dependencies are often constrained within local neighbourhoods in space and time. In particular, we show that for such data, the effective dimension can be far smaller than the ambient dimension, enabling significantly improved convergence rates.

For definitions of the graphs $L_{d \times d'}$ and $L_{d \times d'}^+$, we refer the reader to the examples in Figure 1. While these examples should provide intuitive understanding, formal definitions can be found in Appendix D.

**Images** We begin with the most compelling setting, corresponding to images:

**Lemma 4.4.** *Let $L_{d \times d'}$ be a $d \times d'$ grid graph with $t < d, d'$. The size of the largest clique in $L_{d \times d'}^t$ is less than or equal to $\frac{t^2 + 4t + 3}{2}$.*

**Lemma 4.5.** *Let $L_{d \times d'}^+$ be the $d \times d'$ grid graph with diagonals, and $t < d, d'$. The size of the largest clique in the graph $\left(L_{d \times d'}^+\right)^t$ is $(t+1)^2$.*

Judging from the example in Figure 5, $t = 2$ already gives a fairly reasonable model for images. Thus we have the following dimension-independent rate:

**Corollary 4.6** (Dimension-independent rates)**.** *The neural density estimator in Theorem 4.2 achieves a rate of*

$$\|p - \hat{p}_n\|_1 \in \widetilde{O}_p\left(n^{-1/11.5}\right)$$

*for the grid graph $L_{d \times d'}^2$, and*

$$\|p - \hat{p}_n\|_1 \in \widetilde{O}_p\left(n^{-1/13}\right)$$

*for the grid with diagonals graph* $\left(L_{d \times d'}^{+}\right)^{2}$.

Even when $t > 1$, we have $(t+1)^2 + 2 \ll d$. In practice we expect $t = O(1)$, so even with $t > 1$, the rates are still dimension-independent.

Recall that if a density $p$ is an MRF with respect to a graph $\mathcal{G} = (V, E)$, it is also an MRF with respect to any graph $\mathcal{G}' = (V, E')$ that contains all the edges from $\mathcal{G}$, i.e., $E \subseteq E'$. Thus, the *absence* of edges in an MRF represents a stronger condition on $p$. In the graph $L_{d \times d'}^t$, every $(t+1) \times (t+1)$ block of vertices is fully connected. As demonstrated in Figure 5, when conditioned on an adjacent pixel, pixels tend to become independent with very little distance between them. Figure 5 shows that pixels (8,8) and (9,12) are seemingly independent conditioned on (9,8). Modeling CIFAR-10 as an MRF graph $L_{32 \times 32}^{+}$ would imply that (8,8) and (9,12) are independent conditioned on *every pixel surrounding (8,8)*, a much more stringent requirement than conditioning on one adjacent pixel. Thus, modeling CIFAR-10 as $(L_{32 \times 32}^{+})^2$ appears to be a conservative approach. Consequently, the effective dimension for estimating CIFAR-10 is $(2+1)^2 + 2 = 11$ rather than $32 \times 32 = 1024$, an almost 100-fold improvement!

**Sequences**   For sequential data, we have the following lemma:

**Lemma 4.7.** *Let $L_d$ be a $d$-length path graph. The size of the largest clique in $L_d^t$ is equal to* $\min(t+1, d)$.

Again, we observe that the effective dimension for the neural density estimator, $t+3$, can be far less than the ambient dimension for sequential data, such as audio.

The MRF approach can be extended to various data types, yielding similar dimension reduction results. For instance, color images can be modeled as a three-dimensional random tensor $\mathbf{X} \in \mathbb{R}^{c \times w \times h}$ with a graph $\mathcal{G}$. In this model, the vertices in $\mathbf{X}_{:,i,j} \cup \mathbf{X}_{:,i',j'}$ are fully connected for $|i-i'| \leq 1$ and $|j-j'| \leq 1$, corresponding to a grid graph with diagonals where all channels are connected. Video data can be represented by four-dimensional graphs corresponding to order-4 tensors in $\mathbb{R}^{t \times c \times w \times h}$, with a similar connectivity structure. While text data is discrete in nature, once tokenized and passed through $d$-dimensional word embeddings, it resembles spatial data with dimensions $\mathbb{R}^{d \times t}$ and can benefit from independence structure.

In all these cases, the maximum clique size is determined by how quickly independence is achieved spatio-temporally or in the embedding space, rather than by the overall data dimensionality. This approach yields effective dimensions that are orders of magnitude smaller than the ambient dimension, leading to dimension-independent learning rates.

Crucially, this dimension independence is maintained across varying data sizes. For instance, cropping an image would leave the maximum clique size unchanged (provided the cropping isn't too extreme), while expanding an image would create a larger graph but, assuming the underlying pattern holds, the maximum clique size would remain constant. This property results in a dimension-independent rate of learning that remains consistent across different image sizes. Thus, whether dealing with a $100 \times 100$ pixel image or a $1000 \times 1000$ pixel image of similar content, the effective learning rate remains tied to the maximum clique size rather than the total number of pixels, exemplifying true dimension independence in the learning process.

These extensions demonstrate the versatility of the MRF approach in modeling complex, high-dimensional data structures across various modalities, while significantly reducing the effective dimensionality of the problem.

**Hierarchical models**   Although not the primary focus of this work, our results have potential applications to other data types not typically associated with deep learning. For instance, hierarchical data is often modeled as a rooted tree. For tree-structured MRFs, the following is a well-known:

**Lemma 4.8.** *Let $\mathcal{G}$ be a tree with at least two vertices. The size of the largest clique in $\mathcal{G}$ is 2.*

Estimating densities with a tree MRF has been studied previously: In Liu et al. (2011); Györfi et al. (2022) it was found that one can estimate a density with an unknown tree MRF, without the strong density assumption at a rate $O(n^{-1/4})$. Compared to Theorem 4.2, this is an improvement by a factor of $n^2$, but these estimators are not based on neural networks, which is our focus.

## 5. Conclusion

Neural density estimation has been the subject of intense study over the past few decades, dating at least back to (Magdon-Ismail & Atiya, 1998). There has recently been interest in designing structured neural density estimators that exploit graphical structure (Germain et al., 2015; Johnson et al., 2016; Khemakhem et al., 2021; Wehenkel & Louppe, 2021; Chen et al., 2024). In this work, we have presented a novel perspective on the success of neural networks in density estimation problems. Our approach, based on Markov random fields, offers an alternative explanation to the widely accepted manifold hypothesis for how and why deep learning can circumvent the curse of dimensionality, and aligns with these recent developments on structured density estimation.

We have demonstrated that structured densities that satisfy the Markov property can achieve dimension-independent convergence rates for neural density estimation on spatio-temporal data. Our MRF-based approach complements,

rather than replaces, the manifold hypothesis. We envision a combination of local manifold-like structures and global MRF-like independence properties at play in real-world scenarios with spatio-temporal data, with the manifold hypothesis explaining local features and our MRF approach capturing broader independence structures.

This work opens avenues for future research, including investigating the interplay between local manifold structures and global MRF properties, and developing practical algorithms exploiting these structures within existing models for neural networks.

## Impact Statement

This work provides a theoretical explanation for how neural networks can learn high-dimensional spatio-temporal data with sample-efficient rates. We see no direct societal or ethical risks specific to these results.

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

# A. Proof of Proposition 2.1

Our proof of Proposition 2.1 makes use of the notion of *Hausdorff dimension*, sometimes referred to as the *Hausdorff–Besicovitch dimension*, along with a few of its basic properties. The Hausdorff dimension provides a way to assign a notion of dimension to sets in Euclidean space; for example, a $d'$-dimensional linear subspace of $\mathbb{R}^d$ has Hausdorff dimension $d'$. Before defining the Hausdorff dimension, we first introduce the concept of *Hausdorff measure*. For a measurable set $A \subset \mathbb{R}^d$, let $|A|$ denote its Lebesgue measure.

For any set $A \subset \mathbb{R}^d$, its $s$-dimensional Hausdorff measure is defined as

$$\mathcal{H}^s(A) = \lim_{\delta \to 0} \inf \left\{ \sum_{i=1}^{\infty} |U_i|^s : \{U_i\} \text{ is a } \delta\text{-cover of } A \right\}.$$

The Hausdorff measure is, in some sense, the *natural* notion of measure for sets that may not have the same dimensionality as the ambient space. For example, for all $d$ there exists a constant $c_d > 0$ (the volume of the $d$-dimensional unit ball) such that for all measurable sets $A \subset \mathbb{R}^d$, $c_d \mathcal{H}^d(A) = |A|$ (see equation (2.4) in Falconer (2003)). It also aligns with volume on smooth lower-dimensional subsets, such as manifolds. A set $A$ has finite, positive Hausdorff measure for at most one value of $s$, which is defined to be its Hausdorff dimension. This notion can be extended to sets for which no such value of $s$ exists.

For a set $A \subset \mathbb{R}^d$, its Hausdorff dimension is defined as

$$\dim_{\mathrm{H}}(A) = \inf\{s \geq 0 : \mathcal{H}^s(A) = 0\} = \sup\{s \geq 0 : \mathcal{H}^s(A) = \infty\}.$$

We will not explore Hausdorff dimension further, except to note the following three properties:

1. If $A \subseteq B$, then $\dim_{\mathrm{H}}(A) \leq \dim_{\mathrm{H}}(B)$ (Falconer, 2003, p. 32).

2. If $f : \mathbb{R}^d \to \mathbb{R}^{d'}$ is Lipschitz continuous, then $\dim_{\mathrm{H}}(f(A)) \leq \dim_{\mathrm{H}}(A)$ (Falconer, 2003, Corollary 2.4).

3. If $M$ is a differentiable $m$-dimensional regular submanifold of $\mathbb{R}^d$, then $\dim_{\mathrm{H}}(M) = m$ (Falconer, 2003, p. 32).

*Proof of Proposition 2.1.* We proceed by contradiction. Assume that the proposition setting holds except there exists a subset $X_{I_1}, \ldots, X_{I_{d'+1}}$ that are mutually independent. Without loss of generality, we assume $I_i = i$ for all $i$. Let $\pi : M \to \mathbb{R}^{d'+1}$ be the projection onto the first $d' + 1$ coordinates. Then the pushforward measure $\pi_{\#}\mu$ is given by $\bigotimes_{i=1}^{d'+1} p_i$, and hence $\mathrm{supp}(\pi_{\#}\mu)$ has positive Lebesgue measure in $\mathbb{R}^{d'+1}$. It follows that its Hausdorff dimension is $\dim_{\mathrm{H}}(\mathrm{supp}(\pi_{\#}\mu)) = d' + 1$.

On the other hand, we know that $\mathrm{supp}(\pi_{\#}\mu) \subseteq \pi(M)$ so

$$\dim_{\mathrm{H}}(\mathrm{supp}(\pi_{\#}\mu)) \leq \dim_{\mathrm{H}}(\pi(M)). \tag{2}$$

Since $\pi$ is Lipschitz continuous we have that

$$\dim_{\mathrm{H}}(\pi(M)) \leq \dim_{\mathrm{H}}(M). \tag{3}$$

Finally, since $M$ is a $d'$-dimensional regular submanifold of $\mathbb{R}^d$, we have that

$$\dim_{\mathrm{H}}(M) = d'. \tag{4}$$

Combining equation 2, equation 3, and equation 4 it follows that

$$\dim_{\mathrm{H}}(\mathrm{supp}(\pi_{\#}\mu)) \leq d',$$

which contradictions $\dim_{\mathrm{H}}(\mathrm{supp}(\pi_{\#}\mu)) = d' + 1$ from before. $\qquad\square$

# B. Main Result: Notations and Preliminaries

Before proving the main theorem we will first establish some notation and auxiliary results. For a pair of functions $f, g : \mathbb{X} \to \mathbb{R}$ where $\mathbb{X}$ is an arbitrary domain, we define the $f \cdot g$ to be pointwise function multiplication so $(f \cdot g)(x) = f(x)g(x)$ for all $x \in \mathbb{X}$. For a tuple of functions $f_1, \ldots, f_m : \mathbb{X} \to \mathbb{R}$, the product symbol $\prod_{i=1}^{m} f_i$ is defined to be pointwise function multiplication, i.e., $f_1(x) \cdot f_2(x) \cdot \cdots \cdot f_m(x)$ for all $x \in \mathbb{X}$. Let $\mathbb{N}$ be the set of positive integers. For any $d \in \mathbb{N}$, let $[d] = \{1, 2, \ldots, d\}$.

For a set $V \subset [d]$ with $V = \{v_1, \ldots, v_{|V|}\}$ where $v_i < v_j$ for all $i < j$, let $e_{d,V} : \mathbb{R}^d \to \mathbb{R}^{|V|}; x \mapsto [x_{v_1}, \ldots, x_{v_{|V|}}]$, i.e., $e_{d,V}$ accepts a $d$-dimensional vector and outputs the indices at $V$, in order. The function $e_{V,d}$ can be thought of as selecting some indices from a vector. As a slight abuse of notation, the $d$ subscript will be omitted.

For a graph $\mathcal{G} = (V, E)$, the set of maximal cliques in $\mathcal{G}$ will be denoted $\mathcal{C}(\mathcal{G})$, and is a set of subsets of $V$.

**All of our results will assume the domain of the data is the unit cube** $[0, 1]^d$**.** A density $p$ will be called *positive* if $p(x) > 0$ for all $x \in [0, 1]^d$. Since $[0, 1]^d$ is compact, a direct consequence of this is that there exists $c > 0$ such that $p(x) > c$ for all $x$.

## B.1. Preliminary Results

**Proposition B.1.** *Let $p$ be a Lipschitz continuous probability density $[0, 1]^d$, which is everywhere positive on $[0, 1]^d$ and satisfies the Markov property with respect to a graph $\mathcal{G} = (V, E)$. Then, for all $x \in [0, 1]^d$,*

$$p(x) = \prod_{V' \in \mathcal{C}(\mathcal{G})} \psi_{V'}(e_{V'}(x)),$$

*where each $\psi_{V'}$ is Lipschitz continuous, and there exist constants $c, C$ such that $0 < c \leq C$ and $c \leq \psi_{V'} \leq C$ for all $V' \in \mathcal{C}(\mathcal{G})$.*

Before proving this proposition we first prove the following support lemma.

**Lemma B.2.** *Let $f, g : [0, 1]^d \to \mathbb{R}$ be Lipschitz continuous with $f \geq \delta$ and $g \geq \delta$ for some $\delta > 0$. Then $f \cdot g$ and $1/f$ are both Lipschitz continuous and there exists $\delta' > 0$ such that $f \cdot g \geq \delta'$ and $1/f \geq \delta'$.*

*Proof of Lemma B.2.* Let $f$ be $L_f$-Lipschitz and $g$ be $L_g$-Lipschitz. Because $f$ and $g$ are Lipschitz on a bounded set there exists $C_f > 0$ and $C_g > 0$ such that $f \leq C_f$ and $g \leq C_g$. Let $x, y \in [0, 1]^d$ be arbitrary.

We will begin by proving the product portion of the lemma:

$$\begin{aligned}
|f(x)g(x) - f(y)g(y)| &\leq |f(x)g(x) - f(x)g(y)| + |f(x)g(y) - f(y)g(y)| \\
&\leq C_f |g(x) - g(y)| + C_g |f(x) - f(y)| \\
&\leq C_f L_g \|x - y\|_2 + C_g L_f \|x - y\|_2 \\
&\leq 2 \max (C_f L_g, C_g L_f) \|x - y\|_2.
\end{aligned}$$

The existence of a positive lower bound for $f \cdot g$ follows immediately from $f \cdot g \geq \delta^2$.

To prove the reciprocal portion of the lemma, observe that the function $x \mapsto 1/x$ is Lipschitz on the range of $f$. Since the composition of Lipschitz functions is itself Lipschitz, it follows that $1/f$ is Lipschitz. Finally we have that $1/f \geq 1/C_f > 0$, finishing the proof. $\square$

*Proof of Proposition B.1.* This proof utilizes results from Chang (2007), a work-in-progress book currently used primarily as lecture notes. This work has a constructive proof of the Hammersley-Clifford Theorem. For $S \subset [d]$, let $\gamma_S : \mathbb{R}^d \to \mathbb{R}^d; x \mapsto [x_i \mathbf{1}(i \in S)]_i$, i.e., indices outside of $S$ are set to zero (the zero vector could actually be set to any arbitrary but fixed vector, e.g., set a vector $y \in \mathbb{R}^d$ and $[\gamma_S]_{i \notin S} = y_i$). From the Chang (2007) (3.15), the proof of the Hammersley-Clifford Theorem, it is shown that:

$$p(x) = \prod_{V' \in \mathcal{C}(\mathcal{G})} \psi_{V'}(x),$$

where,

$$\psi_{V'}(x) = \frac{\prod_{V'' \subset V' : |V' \setminus V''| \mod 2 = 0} p\left(\gamma_{V''}(x)\right)}{\prod_{V'' \subset V' : |V' \setminus V''| \mod 2 = 1} p\left(\gamma_{V''}(x)\right)}. \tag{5}$$

For clarity we do an example of the index set of the product; so

$$V'' \subset V' : |V' \setminus V''| \mod 2 = 0$$

denotes a product over all subsets of $V'$ where the set $V''$ where $V'' \cap V'^C$ contains an even number of elements. From equation 5 it is clear that $\psi_{V'}$ only depends on the indices of $x$ in $V'$. The regularity conditions hold due to Lemma B.2.

$\square$

Given any set $S$. For any subset $S' \subseteq S$, let $\mathbf{1}_{S'}$ be the indicator function from $S$ to $\{0, 1\}$, i.e.

$$\mathbf{1}_{S'}(x) = \begin{cases} 0 & \text{if } x \notin S' \\ 1 & \text{if } x \in S' \end{cases} \quad \text{for all } x \in S.$$

Given any $d, b \in \mathbb{N}$, $V = \{v_1, \ldots, v_{|V|}\} \subset [d]$ and $C \geq 1$. For any $A \in [b]^{|V|}$, let $\Lambda_{d,b,A,V}$ be the subset of $[0,1]^d$

$$\Lambda_{d,b,A,V} := \left\{ x \in [0,1]^d \mid x_{v_i} \in \left[ \frac{A_i - 1}{b}, \frac{A_i}{b} \right] \quad \text{for all } i \in [|V|] \right\}. \tag{6}$$

Let $\mathbb{Q}_{d,b,V,C}$ be the set of functions from $[0,1]^d \to \mathbb{R}$

$$\mathbb{Q}_{d,b,V,C} := \left\{ x \mapsto \sum_{A \in [b]^{|V|}} w_A \mathbf{1}_{\Lambda_{d,b,A,V}}(x) \mid w_A \in [0,C] \right\}. \tag{7}$$

For a set $\mathcal{L} \subset L^\beta\left([0,1]^d\right)$ where $1 \leq \beta < \infty$ and $\epsilon > 0$, a subset $\mathcal{C} \subseteq \mathcal{L}$ is called an $\epsilon$-cover of $\mathcal{L}$ in $L^\beta$ norm if, for any $f \in \mathcal{L}$, there exists a $g \in \mathcal{C}$ such that $\|f - g\|_\beta \leq \epsilon$. Also, we define $N(\mathcal{L}, \epsilon)$ to be the cardinality of the smallest subset of $\mathcal{L}$ that is a (closed) $\epsilon$-cover of $\mathcal{L}$ in $L^\beta$ norm. Note that $N(\mathcal{L}, \epsilon)$ depends on $\beta$. We will not specify it when it is clear in the context.

## C. Proof of Theorem 4.2

**Theorem C.1.** *Let $\mathcal{G} = (V, E)$ be a finite graph and $r$ is the size of the largest clique in $\mathcal{G}$. There exists a neural network architecture $\mathcal{F}^*$, such that, for*

$$\hat{p}_n = \arg \min_{f \in \mathcal{F}^*} \|f\|_2^2 - \frac{2}{n} \sum_{i=1}^n f(X_i)$$

*where $X_1, \ldots, X_n \overset{i.i.d.}{\sim} p$, then*

$$\|p - \hat{p}_n\|_2 \in \widetilde{O}_p\left(n^{-1/(4+r)}\right),$$

*for any Lipschitz continuous, positive density $p$ satisfying the Markov property with $\mathcal{G}$.*

This is stronger than $L^1$ convergence since, through Hölder's inequality, we get $L^1$ convergence at the same rate.

**Lemma C.2.** *Let $(\Omega, \Sigma, \mu)$ be a measure space, and let $f_1, \ldots, f_m$ and $g_1, \ldots, g_m$ be measurable and absolutely integrable functions on $\Omega$. Further suppose there exists a constant $C \geq 0$ such that, for all $i \in [m]$,*

$$\|f_i\|_\infty \leq C \quad \text{and} \quad \|g_i\|_\infty \leq C.$$

*Then the following inequality holds:*

$$\left\| \prod_{i=1}^m f_i - \prod_{i=1}^m g_i \right\|_\infty \leq C^{m-1} \sum_{i=1}^m \|f_i - g_i\|_\infty.$$

*Proof of Lemma C.2.* We will proceed by induction on $m$.

**Case** $m = 1$**:** Trivial.

**Induction:** Suppose the lemma holds for some value of $m$. From the inductive hypothesis we have that

$$\left\| \prod_{i=1}^{m+1} f_i - \prod_{i=1}^{m+1} g_i \right\|_\infty \leq \left\| \prod_{i=1}^{m} f_i \cdot f_{m+1} - \prod_{i=1}^{m} g_i \cdot f_{m+1} \right\|_\infty + \left\| \prod_{i=1}^{m} g_i \cdot f_{m+1} - \prod_{i=1}^{m} g_i \cdot g_{m+1} \right\|_\infty$$

$$\leq \left\| \prod_{i=1}^{m} f_i - \prod_{i=1}^{m} g_i \right\|_\infty \|f_{m+1}\|_\infty + \left\| \prod_{i=1}^{m} g_i \right\|_\infty \|f_{m+1} - g_{m+1}\|_\infty$$

$$\leq \left\| \prod_{i=1}^{m} f_i - \prod_{i=1}^{m} g_i \right\|_\infty C + C^m \|f_{m+1} - g_{m+1}\|_\infty$$

$$\leq C^{m-1} \sum_{i=1}^{m} \|f_i - g_i\|_\infty C + C^m \|f_{m+1} - g_{m+1}\|_\infty$$

$$\leq C^m \sum_{i=1}^{m+1} \|f_i - g_i\|_\infty.$$

$\square$

**Space of Neural Network Architectures** Define a space of neural networks as follows. Let $\sigma$ be the ReLU activation function with will act element-wise on vectors. For any $\ell \in \mathbb{N}$, $w = (w_0, \dots, w_{\ell+1})$ with $w_i \in \mathbb{N}$, $s \in \mathbb{N}$ and $F > 0$, the space $\mathcal{F}(\ell, w, s, F)$ is defined by the functions $f : [0,1]^{w_0} \to \mathbb{R}^{w_{\ell+1}}$ which have the form:

$$f(x) = W_\ell \sigma_{v_\ell} W_{\ell-1} \sigma_{v_{\ell-1}} \cdots W_1 \sigma_{v_1} W_0 x,$$

where $\sigma_{v_i}(y) = \sigma(y - v_i)$, $W_i \in \mathbb{R}^{w_{i+1} \times w_i}$, where every entry in $W_i$ and $v_i$ have absolute value less than or equal to 1, $\|f\|_\infty \leq F$, and sum of the total number of nonzero entries of $W_i$ and $v_i$ is less than or equal to $s$. In this work the output dimension of all neural networks will be 1, i.e. $w_{\ell+1}$ will always be assumed to be 1. This is the same space of neural network models employed by (Schmidt-Hieber, 2017).

**Theorem C.3** (Theorem 5, Schmidt-Hieber, 2017). *For any $f \in C_d^\beta([0,1]^d, K)$ and any integers $m \geq 1$ and $N \geq \max((\beta+1)^d, (K+1)e^d)$, there exists a ReLU network $\tilde{f} \in \mathcal{F}(\ell, w, s, \infty)$ with depth*

$$\ell = 8 + (m+5)(1 + \lceil \log_2(\max(d, \beta)) \rceil), \tag{8}$$

*widths*

$$w = (d, 6(d + \lceil \beta \rceil)N, \dots, 6(d + \lceil \beta \rceil)N, 1), \tag{9}$$

*and sparsity*

$$s \leq 141(d + \beta + 1)^{d+3}N(m+6) \tag{10}$$

*such that*

$$\left\| \tilde{f} - f \right\|_{L^\infty([0,1]^d)} \leq (2K+1)(1 + d^2 + \beta^2)6^d N 2^{-m} + K 3^\beta N^{-\beta/d}. \tag{11}$$

**Lemma C.4** (Lemma 5, Remark 1, Schmidt-Hieber, 2017). *For any $\epsilon > 0$,*

$$\log N(\mathcal{F}(\ell, w, s, \infty), \epsilon, \|\cdot\|_\infty) \leq (s+1) \log(2^{2\ell+5} \epsilon^{-1} (\ell+1) w_0^2 w_{\ell+1}^2 s^{2\ell}).$$

### C.1. Main Estimator Proofs

*Proof of Theorem C.1.* Recall that, given a graph $\mathcal{G}$, $\mathcal{C}(\mathcal{G})$ is the set of maximal cliques in $\mathcal{G}$. For any $V' \in \mathcal{C}(\mathcal{G})$, let $\mathcal{F}_{V'} = \mathcal{F}(\ell_{V'}, w_{V'}, s, C)$ where $\ell_{V'}, w_{V'}, s, C$ will be determined later. Also, let

$$\mathcal{F}^* = \left\{ \prod_{V' \in \mathcal{C}(\mathcal{G})} q_{V'} \circ e_{V'} \,\middle|\, q_{V'} \in \mathcal{F}_{V'} \right\}. \tag{12}$$

We shall show that $\mathcal{F}^*$ is the neural network architecture satisfying the desired guarantees in Theorem C.1.

For any set of $n$ i.i.d. samples $X_1, \ldots, X_n$ drawn from $p$, let

$$p_n^* = \arg \min_{f \in \mathcal{F}^*} \|p - f\|_2^2 \quad \text{and} \quad \hat{p}_n = \arg \min_{f \in \mathcal{F}^*} \left( \|f\|_2^2 - \frac{2}{n} \sum_{i=1}^{n} f(X_i) \right). \tag{13}$$

Now, we would like to bound the term $\|\hat{p}_n - p\|_2^2$. We first express it as

$$\|\hat{p}_n - p\|_2^2 = \left( \|\hat{p}_n - p\|_2^2 - \|p_n^* - p\|_2^2 \right) + \|p_n^* - p\|_2^2.$$

For the term $\|\hat{p}_n - p\|_2^2 - \|p_n^* - p\|_2^2$, we further express it as

$$\|\hat{p}_n - p\|_2^2 - \|p_n^* - p\|_2^2 = \underbrace{\|\hat{p}_n - p\|_2^2 - \left( \|p\|_2^2 + \|\hat{p}_n\|_2^2 - \frac{2}{n} \sum_{i=1}^{n} \hat{p}_n(X_i) \right)}_{:=A}$$

$$+ \underbrace{\left( \|p\|_2^2 + \|\hat{p}_n\|_2^2 - \frac{2}{n} \sum_{i=1}^{n} \hat{p}_n(X_i) \right) - \|p_n^* - p\|_2^2}_{:=B} \tag{14}$$

Before we bound $A$ and $B$, we first provide a useful inequality. For any $p' \in \mathcal{F}^*$, we have

$$\|p' - p\|_2^2 - \left( \|p\|_2^2 + \|p'\|_2^2 - \frac{2}{n} \sum_{i=1}^{n} p'(X_i) \right)$$

$$= \frac{2}{n} \sum_{i=1}^{n} p'(X_i) - 2 \langle p', p \rangle$$

$$\leq 2 \max_{f \in \mathcal{F}^*} \left| \mathbb{E}_p(f) - \frac{1}{n} \sum_{i=1}^{n} f(X_i) \right| \quad \text{since } \langle p', p \rangle = \mathbb{E}_p(p') \text{ and } p' \in \mathcal{F}^*. \tag{15}$$

For the term $A$, we immediately have

$$A = \|\hat{p}_n - p\|_2^2 - \left( \|p\|_2^2 + \|\hat{p}_n\|_2^2 - \frac{2}{n} \sum_{i=1}^{n} \hat{p}_n(X_i) \right)$$

$$\leq 2 \max_{f \in \mathcal{F}^*} \left| \mathbb{E}_p(f) - \frac{1}{n} \sum_{i=1}^{n} f(X_i) \right| \quad \text{since } \hat{p}_n \in \mathcal{F}^*.$$

For the term $B$, we have

$$B = \left( \|p\|_2^2 + \|\hat{p}_n\|_2^2 - \frac{2}{n} \sum_{i=1}^{n} \hat{p}_n(X_i) \right) - \|p_n^* - p\|_2^2$$

$$\leq \left( \|p\|_2^2 + \|p_n^*\|_2^2 - \frac{2}{n} \sum_{i=1}^{n} p_n^*(X_i) \right) - \|p_n^* - p\|_2^2 \quad \text{by the optimality of } \hat{p}_n$$

$$\leq 2 \max_{f \in \mathcal{F}^*} \left| \mathbb{E}_p(f) - \frac{1}{n} \sum_{i=1}^{n} f(X_i) \right| \quad \text{since } \hat{p}_n \in \mathcal{F}^*.$$

By plugging them into equation 14, we have

$$\|\hat{p}_n - p\|_2^2 \leq 4 \max_{f \in \mathcal{F}^*} \left| \mathbb{E}_p(f) - \frac{1}{n} \sum_{i=1}^{n} f(X_i) \right| + \|p_n^* - p\|_2^2. \tag{16}$$

We first analyze the term $\|p_n^* - p\|_2^2$ in equation 16. From Proposition B.1, we have that

$$p = \prod_{V' \in \mathcal{C}(\mathcal{G})} \psi_{V'} \circ e_{V'}$$

and there exists some $C_\psi > 0$ so that $\psi_{V'} \leq C_\psi$ for all $V'$ and that, for some $L_\psi$, all $\psi_{V'}$ are $L_\psi$-Lipschitz continuous. We pick a sufficiently large $C$ that is greater than $C_\psi$. Also, by the definition of $\mathcal{F}^*$ in equation 12, we can pick a $q_{V'} \in \mathcal{F}_{V'}$ for each $V' \in \mathcal{C}(\mathcal{G})$ and form an $f \in \mathcal{F}^*$ such that

$$f = \prod_{V' \in \mathcal{C}(\mathcal{G})} q_{V'} \circ e_{V'}.$$

We will specify each $q_{V'}$ later. Then, we have

$$\|f - p\|_\infty = \left\| \prod_{V' \in \mathcal{C}(\mathcal{G})} q_{V'} \circ e_{V'} - \prod_{V' \in \mathcal{C}(\mathcal{G})} \psi_{V'} \circ e_{V'} \right\|_\infty$$

$$\leq C^{|\mathcal{C}(\mathcal{G})|-1} \sum_{V' \in \mathcal{C}(\mathcal{G})} \left\| q_{V'} \circ e_{V'} - \psi_{V'} \circ e_{V'} \right\|_\infty \qquad \text{by Lemma C.2} \qquad (17)$$

Recall that $\mathcal{F}_{V'} = \mathcal{F}(\ell_{V'}, w_{V'}, s, C)$. For any sufficiently large $m, N \in \mathbb{N}$ which we will determine later, we pick

$$\ell_{V'} = 8 + (m+5)(1 + \lceil \log_2 |V'| \rceil),$$
$$w_{V'} = (|V'|, 6(|V'|+1)N, 6(|V'|+1)N, \ldots, 6(|V'|+1)N, 1),$$
$$s = \lfloor 141(r+2)^{r+3} N(m+6) \rfloor$$

and recall that we have picked $C$ to be a constant larger than $C_\psi$ before. It is easy to check that the hypotheses of Theorem C.3 are satisfied with $K = L_\psi$, $\beta = 1$ and $d = |V'|$ and hence, by Theorem C.3, if we pick

$$q_{V'} = \arg \min_{q'_{V'} \in \mathcal{F}_{V'}} \left\| q'_{V'} \circ e_{V'} - \psi_{V'} \circ e_{V'} \right\|_\infty$$

then we have

$$\left\| q_{V'} \circ e_{V'} - \psi_{V'} \circ e_{V'} \right\|_\infty \leq (2L_\psi + 1)(1 + |V'|^2 + 1)6^{|V'|} N 2^{-m} + L_\psi 3 N^{-1/|V'|}$$

$$= O(N 2^{-m} + N^{-1/r}) \qquad (18)$$

By plugging equation 18 into equation 17, we have

$$\|f - p\|_\infty \leq C^{|\mathcal{C}(\mathcal{G})|-1} \sum_{V' \in \mathcal{C}(\mathcal{G})} O(N 2^{-m} + N^{-1/r}) = O(N 2^{-m} + N^{-1/r})$$

Recall that the domain is $[0,1]^d$ and hence we have

$$\|f - p\|_2^2 = \int_{[0,1]^d} |f(x) - p(x)|^2 dx \leq \|f - p\|_\infty^2.$$

Now, by the optimality of $p_n^*$ in equation 13, we have

$$\|p_n^* - p\|_2^2 \leq \|f - p\|_2^2 \leq \|f - p\|_\infty^2 = O(N^2 2^{-2m} + N^{-2/r}). \qquad (19)$$

Now, we take care of the term $\max_{f \in \mathcal{F}^*} \left| \mathbb{E}_p(f) - \frac{1}{n} \sum_{i=1}^n f(X_i) \right|$ in equation 16. To bound this term for all $f \in \mathcal{F}^*$, we first construct an $\epsilon$-cover of $\mathcal{F}^*$ in $L^\infty$. Then, we use the Hoeffding's inequality to bound this term for each $f$ in the $\epsilon$-cover

and use the union bound to control the total failure probability. To construct an $\epsilon$-cover, we define the following notations. For any $V' \in \mathcal{C}(\mathcal{G})$, let $\widetilde{\mathcal{F}}_{V'}$ be a minimal $\frac{\epsilon}{C^{|\mathcal{C}(\mathcal{G})|-1}}$-cover of $\mathcal{F}_{V'}$ in $L^\infty$ where $\epsilon$ is a sufficiently small value and we will determine it later. Also, let

$$\widetilde{\mathcal{F}}^* = \left\{ \prod_{V' \in \mathcal{C}(\mathcal{G})} \tilde{q}_{V'} \circ e_{V'} \mid \tilde{q}_{V'} \in \widetilde{\mathcal{F}}_{V'} \right\}. \tag{20}$$

We will show that $\widetilde{\mathcal{F}}^*$ is an $\epsilon$-cover of $\mathcal{F}$ in $L^\infty$. For any $f \in \mathcal{F}$, it can be expressed as

$$f = \prod_{V' \in \mathcal{C}(\mathcal{G})} q_{V'} \circ e_{V'} \quad \text{for some } q_{V'} \in Q_{V'}$$

Since $\widetilde{\mathcal{F}}_{V'}$ is an $\frac{\epsilon}{C^{|\mathcal{C}(\mathcal{G})|-1}}$-cover of $\mathcal{F}_{V'}$ in $L^\infty$ for all $V' \in \mathcal{C}(\mathcal{G})$, there exists a $\tilde{q}_{V'} \in \widetilde{\mathcal{F}}_{V'}$ such that

$$\|q_{V'} - \tilde{q}_{V'}\|_\infty \leq \frac{\epsilon}{C^{|\mathcal{C}(\mathcal{G})|-1}}.$$

By the definition of $\widetilde{\mathcal{F}}$, we set $\tilde{f} \in \widetilde{\mathcal{F}}$ to be

$$\tilde{f} = \prod_{V' \in \mathcal{C}(\mathcal{G})} \tilde{q}_{V'} \circ e_{V'}$$

By Lemma C.2, we check that

$$\left\| f - \tilde{f} \right\|_\infty = \left\| \prod_{V' \in \mathcal{C}(\mathcal{G})} q_{V'} - \prod_{V' \in \mathcal{C}(\mathcal{G})} \tilde{q}_{V'} \right\|_\infty = C^{|\mathcal{C}(\mathcal{G})|-1} \cdot \sum_{V' \in \mathcal{C}(\mathcal{G})} \|q_{V'} - \tilde{q}_{V'}\|_\infty$$

$$\leq C^{|\mathcal{C}(\mathcal{G})|-1} \cdot \frac{\epsilon}{C^{|\mathcal{C}(\mathcal{G})|-1}}$$

$$= \epsilon.$$

Now, we return to the term $\max_{f \in \mathcal{F}} \left| \mathbb{E}_p(f) - \frac{1}{n} \sum_{i=1}^n f(X_i) \right|$. Since $\widetilde{\mathcal{F}}^*$ is an $\epsilon$-cover of $\mathcal{F}^*$ in $L^\infty$, for any $f \in \mathcal{F}^*$, there exists a $\tilde{f} \in \widetilde{\mathcal{F}}^*$ such that $\|f - \tilde{f}\|_\infty \leq \epsilon$ and we have

$$\left| \mathbb{E}_p(f) - \frac{1}{n} \sum_{i=1}^n f(X_i) \right|$$

$$\leq \left| \mathbb{E}_p(f) - \mathbb{E}_p(\tilde{f}) \right| + \left| \mathbb{E}_p(\tilde{f}) - \frac{1}{n} \sum_{i=1}^n \tilde{f}(X_i) \right| + \left| \frac{1}{n} \sum_{i=1}^n \tilde{f}(X_i) - \frac{1}{n} \sum_{i=1}^n f(X_i) \right|$$

$$\leq 2\epsilon + \left| \mathbb{E}_p(\tilde{f}) - \frac{1}{n} \sum_{i=1}^n \tilde{f}(X_i) \right|$$

which implies

$$\max_{f \in \mathcal{F}^*} \left| \mathbb{E}_p(f) - \frac{1}{n} \sum_{i=1}^n f(X_i) \right| \leq 2\epsilon + \max_{\tilde{f} \in \widetilde{\mathcal{F}}^*} \left| \mathbb{E}_p(\tilde{f}) - \frac{1}{n} \sum_{i=1}^n \tilde{f}(X_i) \right|. \tag{21}$$

By Hoeffding's inequality and the union bound, for any $t > 0$, the probability of

$$\max_{\tilde{f} \in \widetilde{\mathcal{F}}^*} \left| \mathbb{E}_p(\tilde{f}) - \frac{1}{n} \sum_{i=1}^n \tilde{f}(X_i) \right| > t$$

is bounded by $|\widetilde{\mathcal{F}}^*| \cdot e^{-\Omega(nt^2)}$.

To bound the term $|\widetilde{\mathcal{F}}^*|$, by the definition of $\widetilde{\mathcal{F}}^*$ in equation 20, we first have

$$\log|\widetilde{\mathcal{F}}^*| = \sum_{V' \in \mathcal{C}(\mathcal{G})} \log|\widetilde{\mathcal{F}}_{V'}|.$$

For each term $\log|\widetilde{\mathcal{F}}_{V'}|$, by Lemma C.4, we have

$$\log|\widetilde{\mathcal{F}}_{V'}| \le (s+1)\log(2^{2L_{V'}+5}\epsilon^{-1}(L_{V'}+1)|V'|^2 s^{2L_{V'}}).$$

We now bound the architecture parameters. Recall that

$$\ell_{V'} = 8 + (m+5)(1 + \lceil \log_2 |V'| \rceil) \text{ for any } V' \in \mathcal{C}(\mathcal{G}) \text{ and}$$
$$s = \lfloor 141(r+2)^{r+3}N(m+6) \rfloor.$$

Namely, we have

$$\ell_{V'} = O(m) \quad \text{and} \quad s = O(Nm) \quad \text{which implies} \quad \log|\widetilde{\mathcal{F}}_{V'}| \le O(Nm^2 \log \frac{Nm}{\epsilon}).$$

That means we have

$$\log|\widetilde{\mathcal{F}}^*| \le \sum_{V' \in \mathcal{C}(\mathcal{G})} O(Nm^2 \log \frac{Nm}{\epsilon}) = O(Nm^2 \log \frac{Nm}{\epsilon}).$$

By setting $t = O(\sqrt{\frac{Nm^2}{n} \log \frac{Nnm}{\epsilon}})$, we have

$$\max_{\tilde{f} \in \widetilde{\mathcal{F}}^*} \left| \mathbb{E}_p(\tilde{f}) - \frac{1}{n}\sum_{i=1}^n \tilde{f}(X_i) \right| < O(\sqrt{\frac{Nm^2}{n} \log \frac{Nnm}{\epsilon}})$$

with at least probability $1 - |\widetilde{\mathcal{F}}^*| \cdot e^{-\Omega(nt^2)} \to 1$ as $n \to \infty$. Plugging it into equation 21, we have

$$\max_{f \in \mathcal{F}^*} \left| \mathbb{E}_p(f) - \frac{1}{n}\sum_{i=1}^n f(X_i) \right| \le 2\epsilon + O(\sqrt{\frac{Nm^2}{n} \log \frac{Nnm}{\epsilon}}). \tag{22}$$

Furthermore, by plugging equation 19 and equation 22 into equation 16, we have

$$\|\hat{p}_n - p\|_2^2 \le 4 \max_{f \in Q} \left| \mathbb{E}_p(f) - \frac{1}{n}\sum_{i=1}^n f(X_i) \right| + \|p_n^* - p\|_2^2$$
$$< O(\epsilon + \sqrt{\frac{Nm^2}{n} \log \frac{Nnm}{\epsilon}} + N^2 2^{-2m} + N^{-2/r}).$$

By picking

$$\epsilon = n^{-\frac{2}{r+4}}, \quad N = n^{\frac{r}{r+4}} \quad \text{and} \quad m = \frac{r+1}{r+4}\log n,$$

we have

$$\|\hat{p}_n - p\|_2^2 \le \widetilde{O}(n^{-\frac{2}{r+4}}).$$

$\square$

# D. Graph Proofs

For any $d, d', t \in \mathbb{N}$, define $L_{d \times d'}$ to be the graph whose vertex set is $[d] \times [d']$ and edge set is

$$\{((i,j),(i',j')) \mid i, i' \in [d], j, j' \in [d'], (i,j) \neq (i',j'), |i-j| + |i'-j'| \leq 1\}$$

and $L_{d \times d'}^t$ to be the graph whose vertex set is $[d] \times [d']$ and edge set is

$$\{((i,j),(i',j')) \mid i, i' \in [d], j, j' \in [d'], (i,j) \neq (i',j'), |i-j| + |i'-j'| \leq t\}.$$

For any $d, d', t \in \mathbb{N}$, define $L_{d \times d'}^+$ to be the graph whose vertex set is $[d] \times [d']$ and edge set is

$$\{((i,j),(i',j')) \mid i, i' \in [d], j, j' \in [d'], (i,j) \neq (i',j'), \max\{|i-j|, |i'-j'|\} \leq 1\}$$

and $(L_{d \times d'}^+)^t$ to be the graph whose vertex set is $[d] \times [d']$ and edge set is

$$\{((i,j),(i',j')) \mid i, i' \in [d], j, j' \in [d'], (i,j) \neq (i',j'), \max\{|i-j|, |i'-j'|\} \leq t\}.$$

For any $d, t \in \mathbb{N}$, define $L_d$ to be the graph whose vertex set is $[d]$ and edge set is $\{(i,j) \mid i \neq j, |i-j| \geq 1\}$ and $L_d^t$ to be the graph whose vertex set is $[d]$ and edge set is $\{(i,j) \mid i \neq j, |i-j| \geq t\}$.

*Proof of Lemma 4.4.* For any clique $C$ in $(L_{d \times d'})^t$, let $(i_0, j_0)$ (resp. $(i_1, j_1)$, $(i_0', j_0')$ and $(i_1', j_1')$) be the vertex in $C$ such that $i_0 + j_0$ is maximal (resp. $i_1 + j_1$ is minimal, $i_0' - j_0'$ is maximal and $i_1' - j_1'$ is minimal). Namely, the vertex set of $C$ is a subset of

$$S := \{(i,j) \mid i \in [d], j \in [d'], i_1 + j_1 \leq i + j \leq i_0 + j_0, i_1' - j_1' \leq i - j \leq i_0' - j_0'\}.$$

By the definition of cliques and $(L_{d \times d'})^t$, we have

$$\begin{aligned}(i_0 + j_0) - (i_1 + j_1) &\leq |i_0 - i_1| + |j_0 - j_1| \leq t \quad \text{since there is an edge between } (i_0, j_0) \text{ and } (i_1, j_1) \\ (i_0' - j_0') - (i_1' - j_1') &\leq |i_0' - i_1'| + |j_0' - j_1'| \leq t \quad \text{since there is an edge between } (i_0', j_0') \text{ and } (i_1', j_1')\end{aligned}$$

To bound the size of $S$, we observe that, for each of the at most $t + 1$ possible values $i_1 + j_1, i_1 + j_1 + 1, \ldots, i_0 + j_0$ equal to $i + j$, there are at most $\lceil \frac{t+1}{2} \rceil$ possible values among $i_1' + j_1', i_1' + j_1' + 1, \ldots, i_0' + j_0'$ equal to $i - j$ by considering the parity. Therefore, $|S|$ is at most $(t+1) \cdot \lceil \frac{t+1}{2} \rceil$.

Hence, the size of the largest clique in $(L_{d \times d'})^t$ is at most $(t+1) \cdot \lceil \frac{t+1}{2} \rceil \leq \frac{t^2 + 4t + 3}{2}$. $\qquad \square$

*Proof of Lemma 4.5.* It is easy to check that the subgraph of $(L_{d \times d}^+)^t$ induced by the vertex set $[t+1] \times [t+1]$ is a clique. Hence, the size of the largest clique in $(L_{d \times d'}^+)^t$ is at least $(t+1)^2$.

For any clique $C$ in $(L_{d \times d'}^+)^t$, let $i_0$ (resp. $i_0'$) be the smallest (resp. largest) first index of the vertices in $C$ and $j_0$ (resp. $j_0'$) be the smallest (resp. largest) second index of the vertices in $C$. Namely, the vertex set of $C$ is a subset of

$$S := \{(i,j) \mid i \in [d], j \in [d'], i_0 \leq i \leq i_0', j_0 \leq j \leq j_0'\}.$$

To bound the size of $S$, by the definition of cliques and $(L_{d \times d'}^+)^t$, we have

$$i_0' - i_0 \leq t \quad \text{and} \quad j_0' - j_0 \leq t$$

Therefore, $|S|$ is at most $(t+1)^2$.

Hence, the size of the largest clique in $(L_{d \times d'}^+)^t$ is $(t+1)^2$.

$\qquad \square$

*Proof of Lemma 4.7.* It is easy to check that the subgraph of $L_d^t$ induced by the vertex set $[\min\{t+1, d\}]$ is a clique. Hence, the size of the largest clique in $L_d^t$ is at least $\min\{t+1, d\}$.

For any clique $C$ in $L_d^t$, let $i_0$ (resp. $j_0$) be the smallest (resp. largest) index of the vertex in $C$. Namely, the vertex set of $C$ is be a subset of $S := \{i | i \in [d], i_0 \leq i \leq j_0\}$. By the definition of cliques and $L_d^t$, we have $|i - j| \leq \min\{t, d-1\}$. Therefore, $|S|$ is at most $\min\{t+1, d\}$.

Hence, the size of the largest clique in $L_d^t$ is $\min\{t+1, d\}$. $\qquad\qquad\qquad\qquad\qquad\qquad\qquad$ □

# E. COCO and Google Speech Commands Scatter Plots

**COCO**

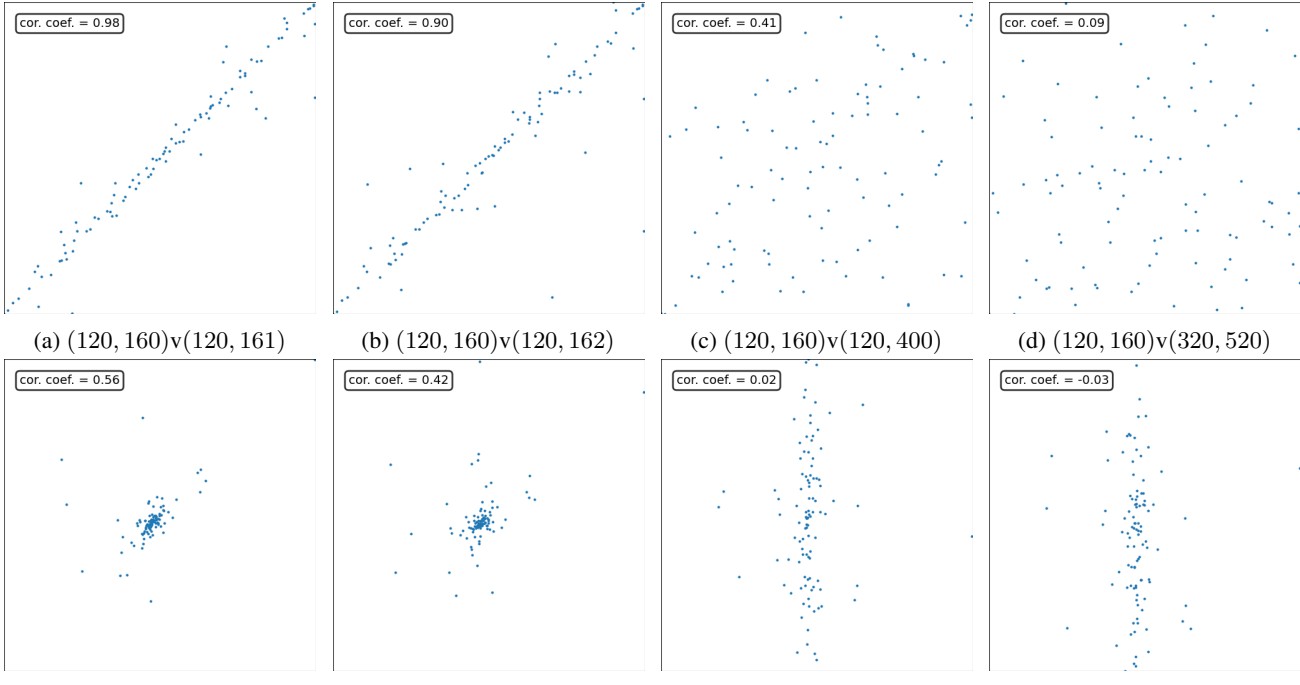

(a) $(120, 160)\mathrm{v}(120, 161)$  (b) $(120, 160)\mathrm{v}(120, 162)$  (c) $(120, 160)\mathrm{v}(120, 400)$  (d) $(120, 160)\mathrm{v}(320, 520)$

(e) $(120, 160)\mathrm{v}(120, 161)$ cond.  (f) $(120, 160)\mathrm{v}(120, 162)$ cond.  (g) $(120, 160)\mathrm{v}(120, 400)$ cond.  (h) $(120, 160)\mathrm{v}(320, 520)$ cond.

Figure 7: This figure presents scatter plots analogous to those in Figure 5 of the main text, but derived from the COCO training set (Lin et al., 2014). The conditional scatter plots are based on pixel (121,160) being near its median value.

Due to memory constraints, we used a subset of the data:

1. 4000 random samples were initially selected.

2. From these, 100 images with pixel (121,160) nearest to the median were chosen for the conditional plots.

Note that increasing the sample size for conditioning resulted in lower observed correlation. This is because a larger sample allows for a more precise conditioning, better approximating the true conditional distribution. The wider the range of values for the conditioning pixel (121,160), the more the selected points resemble the unconditional distribution, potentially introducing spurious correlation.

**Google Speech Commands**

Here we consider the correlations and conditional correlations for covariates from audio data. We use the Google Speech Commands Dataset (Warden, 2018), focusing on the "no" class. The corresponding plots are shown in Figure 8.

We note that the dataset contains more extreme outliers than the image datasets. For example, while 50% of all entries have magnitude 80 or less, the maximum magnitude in the dataset is 32,767. This causes some of the scatterplots in Figure 8 to appear off-center. We use 500 randomly selected samples, which helps reduce visual artifacts from quantization when plotting conditional correlations with fewer samples.

Unlike Figures 5 and 7, the $x$- and $y$-axes in each image here are not normalized to the same scale, and the $x$-values in the conditional plots are often skewed. None of these issues affect the underlying correlation or conditional independence properties we aim to illustrate. While we explored ways to account for these quirks in the visualizations, we ultimately opted to present the raw plots for simplicity, as the effect on dependence structure is still clearly visible.

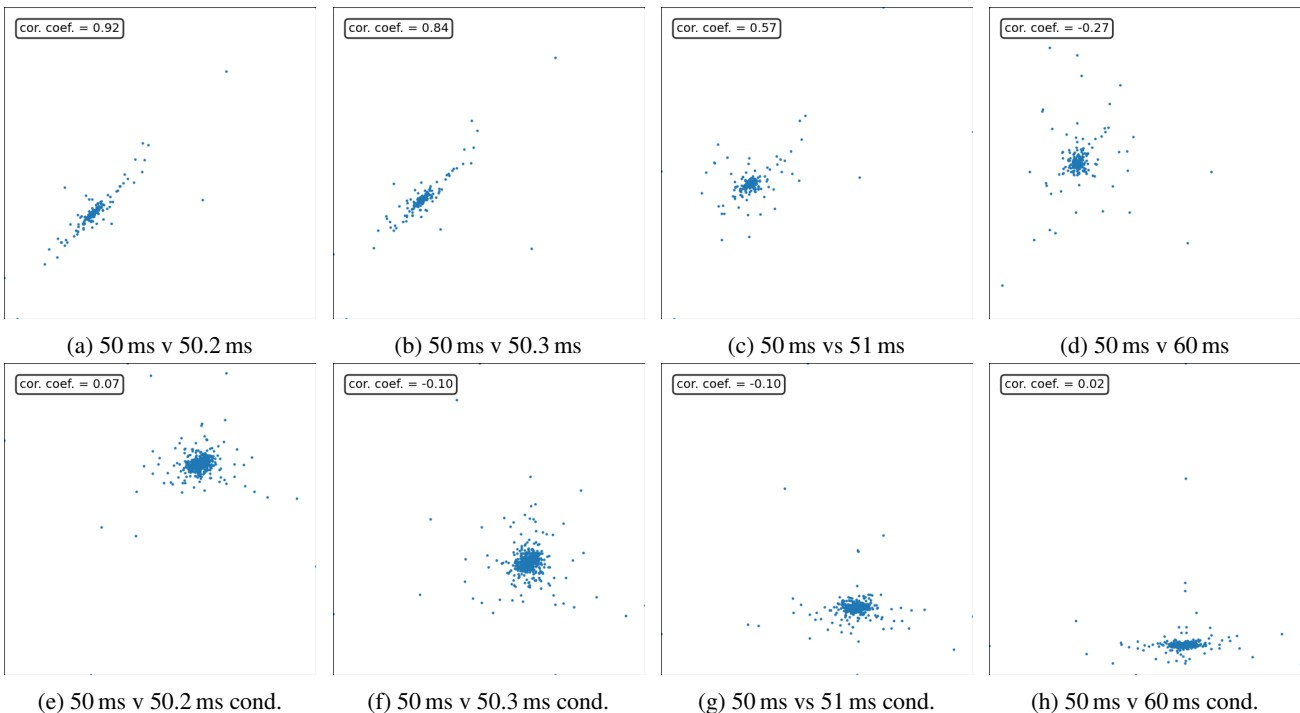

| (a) 50 ms v 50.2 ms | (b) 50 ms v 50.3 ms | (c) 50 ms vs 51 ms | (d) 50 ms v 60 ms |
|---|---|---|---|
| (e) 50 ms v 50.2 ms cond. | (f) 50 ms v 50.3 ms cond. | (g) 50 ms vs 51 ms cond. | (h) 50 ms v 60 ms cond. |

Figure 8: This figure presents scatter plots analogous to those in Figures 5 and 7, but derived from the "no" class of the Google Speech Commands Dataset (Warden, 2018). The plots compare the sample value at 50 ms with the values at other time points. The conditional graphs (bottom row) show the same comparisons, but conditioned on the value of the sample immediately after 50 ms being near its median. Note that the sampling rate of this dataset is 16,000 Hz, so each sample corresponds to approximately 0.1 ms.

## F. Decoupling the Manifold Hypothesis and the Markov Random Field Assumption

Here, we present four simple examples illustrating that the manifold hypothesis and the MRF assumption are, in a fundamental sense, independent modeling concepts.

- MRF true, MH true: $X \sim \mathrm{unif}[0,1]$, $Y \sim \mathrm{unif}[0,0.01]$, $X$ and $Y$ independent

- MRF true, MH false: $X, Y \sim \mathrm{unif}[0,1]$, $X$ and $Y$ independent

- MRF false, MH true: $X \sim \mathrm{unif}[0,1]$, $X = Y$

- MRF false, MH false: $X, Y \sim N(0,1)$, weakly correlated

