# OpenReview forum: "Dimension-Independent Rates for Structured Neural Density Estimation"
_ICML.cc/2025/Conference — ICML 2025 poster_

### Official Review · Reviewer_SQfd · 2025-03-12

**Overall Recommendation:** 3

**Summary:**

The authors study density estimation for the setting of Markov Random Field (MRF) data where we only have local correlations (cliques). Under these assumptions they study the convergence rate of Neural Networks with their bounds only depend on the range of the correlations. Morally the dimension in convergence bounds can be substituted with the dimension of the effective dimension of the correlation range.

**Claims And Evidence:**

The claims are supported by evidence.

**Essential References Not Discussed:**

None that I know of.

**Experimental Designs Or Analyses:**

Experiments seem ok.

**Methods And Evaluation Criteria:**

Yes, given that this is a theory paper.

**Other Comments Or Suggestions:**

- In the abstract who is $d$, it does not appear in the formulas?

**Other Strengths And Weaknesses:**

Strengths:
- Well motivated
- Clearly written
- From the theory side this is interesting research that can be built on.

Weaknesses:
- The main result (Theorem 4.2) is quite abstract in the sense, that it only makes a qualitative statement about the existence of neural networks achieving the bound. This in combination without sufficient practical experiments makes it difficult to judge how impactful this result is.
- In terms of technical novelty it is not too much. The proofs use mostly standard techniques with the main results leaning on ideas from Schmidt-Hieber 17. ( This is critique solely on the *technical* novelty, considering known techniques in a different setting and carrying out all the computations and proofs does provide value)
- - What seems missing to me a bit is how one would go about estimating *r* for a given dataset. Obviously for any real data there is nonzero but potentially negligible correlation based on distance. However it would be good if there was a rigorous way to determine what negligible means and how it is determined.

**Questions For Authors:**

- How would one determine $r$ for a real dataset (efficiently)? What if the correlation is not zero far out but rather very small (think of $\epsilon << 1$)? I am not an expert in that literature but in general most graph problems can be computationally quite expensive.
- On thing I am missing a bit motivation wise is what the practical implications would be from this theorem. Too me it seems that the theorem mostly states that local (i.e. simpler) structure leads to better bound, which in itself is not surprising. What takeaway is there beyond that? In particular do you believe that the rate would be reasonable baseline for practical settings?

**Relation To Broader Scientific Literature:**

These results are relevant to the broader

**Theoretical Claims:**

Checked them high level (i.e. skimmed the proof), did not find any major issue.

---

> ### Author Rebuttal · Authors · 2025-03-31
>
> __“The main result (Theorem 4.2) is quite abstract in the sense, that it only makes a qualitative statement about the existence of neural networks achieving the bound. This in combination without sufficient practical experiments makes it difficult to judge how impactful this result is.”__
>
> We appreciate the reviewer’s concern about the limited empirical validation. While we acknowledge this limitation, we note that:
> * The paper's primary contribution is theoretical: Proving rigorously that neural networks can achieve dimension-independent rates in density estimation under commonly accepted assumptions. This provides yet another compelling justification for using DNNs in practice.
> * The local dependency structure we identify is already implicitly leveraged in successful practical methods, particularly in convolutional architectures and patch-based approaches to anomaly detection such as SoftPatch (NeurIPS 2022). Convolutional structures inherently exploit local dependencies through limited receptive fields, closely aligning with the Hammersley-Clifford theorem, which we use to prove our results.
> * Due to strict space constraints (we are already at the page limit), we focused on developing the theoretical and intuitive foundations thoroughly. A comprehensive empirical study would require significant additional space to properly evaluate different architectures, datasets, and parameter settings.
>
> __“On thing I am missing a bit motivation wise is what the practical implications would be from this theorem. Too me it seems that the theorem mostly states that local (i.e. simpler) structure leads to better bound, which in itself is not surprising. What takeaway is there beyond that? In particular do you believe that the rate would be reasonable baseline for practical settings?”__
>
> The fact that real-world densities can be estimated with (a) dimension-independent rates through (b) neural density estimation is a surprising result, and indeed suggests this presents a reasonable baseline in practical settings. For the reasons outlined above, we leave it to future work to evaluate practical aspects more thoroughly, and indeed, we believe this is an important direction to pursue.
>
> Moreover, the significant attention devoted simply to validating the manifold hypothesis [1,2,3] illustrates the community's interest in identifying structural assumptions that explain learnability in real-world data. From this viewpoint, the MRF framework we propose provides another natural way of suggesting structured function spaces—especially given that the Hammersley-Clifford theorem explicitly characterizes functions reflecting local conditional independence.
>
> __"What seems missing to me a bit is how one would go about estimating $r$ for a given dataset. Obviously for any real data there is nonzero but potentially negligible correlation based on distance. However it would be good if there was a rigorous way to determine what negligible means and how it is determined."__
>
> __“How would one determine $r$ for a real dataset (efficiently)? What if the correlation is not zero far out but rather very small (think of $\epsilon \ll1$)? I am not an expert in that literature but in general most graph problems can be computationally quite expensive.__
>
> A significant advantage of the settings we consider is that they naturally suggest reasonable MRF structures (along with their $r$) that can be verified empirically, as we have shown. Indeed, datatypes for which DNNs have historically shown exceptional success—such as images, audio, and text—naturally exhibit conditional dependence structures closely related to the MRF framework we propose. This connection is well-established, as evidenced by comprehensive textbooks dedicated to applying Markov Random Fields to image processing tasks [4,5] .
>
> Outside of these settings, we acknowledge that identifying the graph or $r$ from data can be challenging. With the extra space allotted for the camera ready we will include related work on the estimation of MRF graphs and $r$.
>
> __“In the abstract who is $d$, it does not appear in the formulas?”__
>
> This was an oversight on our part: $d$ is the ambient dimension of our data, i.e., the dimension of the data whose pdf we are estimating. We will make this clear in the camera ready.
>
> [1] Pope et al. The intrinsic dimension of images and its impact on learning. ICLR, 2021(in paper)
>
> [2] Carlsson et al. On the Local Behavior of Spaces of Natural Images. International Journal of Computer Vision, 76(1):1–12, January 2008.
>
> [3] Brown et al. The Union of Manifolds Hypothesis, NeurIPS, 2022 (in paper)
>
> [4] Li, S. Z. Markov Random Field Modeling in Image Analysis, 2009 (3000+ citations)
>
> [5] Blake, A., Kohli, P., and Rother, C. Markov random fields for vision and image processing. 2011

---

> > ### Comment · Reviewer_SQfd · 2025-04-06
> >
> > I thank the authors for addressing my concerns, I have updated the score in my review accordingly

---

### Official Review · Reviewer_2eJb · 2025-03-13

**Overall Recommendation:** 5

**Summary:**

This paper claims to propose a novel theoretical framework that exploits the data structure using Markov Random Fields (MRFs) to provide a dimension independent converge rate for structured density estimation. It shows that using Markov Random Fields allows capturing local dependency between pixels while considering far-away pixels as nearly independent. It addresses the curse of dimensionality complementary to the manifold hypothesis.

**Claims And Evidence:**

The paper suggests using power graphs to model possible arbitrary long-range dependencies in the data structure, which is supposed to be more realistic compared to path and grid MRFs that only model extremely local correlations. This is motivated through the computation of correlation between pixel values in CIFAR-10 in Figure 3 and in COCO in Appendix D.
The paper proposes convergence rates for ReLU fully connected networks that are applicable to spatial (e.g., images) and sequential data (audio, text), spatiotemporal (e.g., videos), and hierarchical data through the main theoretical result which is the Theorem 4.2.

**Essential References Not Discussed:**

N/A

**Experimental Designs Or Analyses:**

Figure 3 shows a scatter plot that motivates the choice of using Markov random fields to model the data distribution, showing that (a) the correlation decreases with the distance and (b) conditioning on a neighboring pixel decreases the correlation. Furthermore, Figure 4 demonstrates the motivation for using Power Graphs instead of standard grid structures to model image distributions.

**Methods And Evaluation Criteria:**

This paper is mostly theoretical, the evaluation of the approach compared to path and grid MRFs is qualitative with the correlation plots of Figure 3.

**Other Comments Or Suggestions:**

While the theorem claims are applicable to diverse data time (audio, text, image), the illustrations are mostly performed with images. It would strengthen the claims to provide plots similar to Figures 3 and 8 (and possibly, an estimate of $r$) with another data type (e.g., with sequential data such as audio or text).

I would also suggest defining $n$ in the abstract.

**Other Strengths And Weaknesses:**

The paper is very well written and provides a comprehensive overview of the field and the problem setup. The contribution is mostly theoretical with the provided convergence rate of Theorem 4.2, and it is then illustrated with numerous examples.

**Questions For Authors:**

The theoretical result holds for ReLU Networks. Any insights on how it may generalize (or not) to other spaces of networks?

**Relation To Broader Scientific Literature:**

The paper is based on the factorization property for Markov Random Fields Hammersley & Clifford, (1971). This is applied to ReLU neural networks considering Theorem 5 of Schmidt-Hieber, (2017). It also discusses the more general manifold hypothesis (Bengio et al., 2013; Brahma et al., 2016) in section 3.3 and in Appendix E. It would be useful to provide more explanation to the unfamiliar reader for the four examples of section E.

**Theoretical Claims:**

For ReLU fully connected networks, the paper claims to obtain a convergence rate in $O(n^{-1/(4+r)})$ where $n$ is the number of training samples and $r$ is the size of the largest clique in G, where G is the graph structure associated with the data distribution. This graph G can be obtained using the power graph of the original graph that represents the data structure to capture long-range dependencies. This is compared in the argumentation with $O(n^{-1/(2+d)})$, thus providing a large improvement with $r + 2 \ll d$. Furthermore, $r$ is claimed to be mostly independent of the data dimension, thus providing Dimension-Independent Rates that alleviate the curse of dimensionality.

---

> ### Author Rebuttal · Authors · 2025-03-31
>
> __“It would be useful to provide more explanation to the unfamiliar reader for the four examples of section E.”__
>
> We will extend this discussion. In our response to Reviewer ZT7H, we included a more technical explanation of how the manifold hypothesis connects to dependence. We can incorporate a similar discussion in the main text or appendix.
>
> __“While the theorem claims are applicable to diverse data time (audio, text, image), the illustrations are mostly performed with images. It would strengthen the claims to provide plots similar to Figures 3 and 8 (and possibly, an estimate of $r$) with another data type (e.g., with sequential data such as audio or text).”__
>
> We will gladly include a similar plot for a different data type in the next draft. It's worth noting that using Markov random fields to model images is well-established—line 215 cites two textbooks that are entirely on the subject. Our goal was simply to provide images to help the reader grasp the intuition.

---

> > ### Comment · Reviewer_2eJb · 2025-04-02
> >
> > Thank you for your response! However, you did not answer to my question:
> > "The theoretical result holds for ReLU Networks. Any insights on how it may generalize (or not) to other spaces of networks?"

---

> > > ### Author Response · Authors · 2025-04-03
> > >
> > > We apologise for the oversight! This is a great question. Our results are not specific to ReLU networks and can be extended to any class of neural networks for which appropriate approximation bounds are available. For example, our results can be extended to non-ReLU activation by using the results of https://arxiv.org/abs/1906.06903 (applicable to sigmoid, tanh, swish, etc.). We omitted this technical detail since ReLU networks are the most common choice in practice, and the proof remains the same.

---

### Official Review · Reviewer_ae2s · 2025-03-14

**Overall Recommendation:** 1

**Summary:**

This paper analyzes density estimation using a neural network under a conditional independence (MRF) assumption and shows that density estimation is possible with a dimension-independent rate under this assumption.

**Claims And Evidence:**

I don't think there's enough evidence here to support the claim that images and audio have the kind of independence structure assumed. The evidence provided seems to just be that conditioning on some particular pixel for CIFAR images seems to give a plot where the pixel correlation is smaller, which is extremely weak.

**Essential References Not Discussed:**

I'm not familiar enough with the literature here to know for sure, but am skeptical that the results in this paper were not already known.

**Experimental Designs Or Analyses:**

No real experiments

**Methods And Evaluation Criteria:**

There are no real experiments here.

**Other Comments Or Suggestions:**

1) Explain relationship to prior work much more thoroughly
2) Need much more convincing evidence to support the claim that images have the kind of conditional independence structure assumed.
3) Most of the paper needs to be spent on explaining your contribution, not on establishing preliminaries, as is currently the case.

**Other Strengths And Weaknesses:**

Half of this paper is spent on just explaining MRFs, which most people in the ML community are probably already at least somewhat familiar with. Many of the claims about images having this conditional independence structure are not well-supported. I am generally skeptical that the theoretical results are not already known.

**Questions For Authors:**

Please look at comments above.

**Relation To Broader Scientific Literature:**

I'm extremely surprised that the result presented here was not already known, perhaps as a corollary of some stronger result. No real comparison with any prior work is provided.

**Theoretical Claims:**

The theoretical claims are correct.

---

> ### Author Rebuttal · Authors · 2025-03-31
>
> __"There are no real experiments here."__
>
> __"No real experiments"__
>
> We appreciate the reviewer’s concern about the limited empirical validation. While we acknowledge this limitation, we note that:
> * The paper's primary contribution is theoretical: Proving rigorously that neural networks can achieve dimension-independent rates in density estimation under commonly accepted assumptions. This provides yet another compelling justification for using DNNs in practice.
> * The local dependency structure we identify is already implicitly leveraged in successful practical methods, particularly in convolutional architectures and patch-based approaches to anomaly detection such as SoftPatch (NeurIPS 2022). Convolutional structures inherently exploit local dependencies through limited receptive fields, closely aligning with the Hammersley-Clifford theorem, which we use to prove our results.
> * Due to strict space constraints (we are already at the page limit), we focused on developing the theoretical and intuitive foundations thoroughly. A comprehensive empirical study would require significant additional space to properly evaluate different architectures, datasets, and parameter settings.
>
>
> __“I don't think there's enough evidence here to support the claim that images and audio have the kind of independence structure assumed. The evidence provided seems to just be that conditioning on some particular pixel for CIFAR images seems to give a plot where the pixel correlation is smaller, which is extremely weak.”__
>
> __“Many of the claims about images having this conditional independence structure are not well-supported.”__
>
> __"Need much more convincing evidence to support the claim that images have the kind of conditional independence structure assumed."__
>
> This model is extensively well-supported: On line 215 (left) we cite two entire textbooks on the application of Markov random fields to image analysis:
>
> * Li, S. Z. Markov Random Field Modeling in Image Analysis, 2009 (3000+ citations)
> * Blake, A., Kohli, P., and Rother, C. Markov random fields for vision and image processing. 2011
>
> Other textbooks on Markov random fields for machine learning (e.g., A Blake, P Kohli, C Rother, Markov models for pattern recognition: from theory to applications, 2011) contain applications to text and audio data. For example the classic n-gram text model is exactly a Markov random field.
>
> We included the images simply to give the reader an idea of how this model is manifest in real world data.
>
> __“I'm extremely surprised that the result presented here was not already known, perhaps as a corollary of some stronger result. No real comparison with any prior work is provided.”__
>
> __“I'm not familiar enough with the literature here to know for sure, but am skeptical that the results in this paper were not already known.”__
>
> __“I am generally skeptical that the theoretical results are not already known.”__
>
> __"Explain relationship to prior work much more thoroughly"__
>
> To the best of our knowledge estimators that achieve these rates do not exist, let alone that they can be achieved with a neural network using a reasonably computationally tractable loss function. Our literature review is quite extensive, with over 40 citations, including up to date citations from 2023-24. We even discuss alternative approaches to obtaining faster rates at lines 125-143.
>
> __“Half of this paper is spent on just explaining MRFs, which most people in the ML community are probably already at least somewhat familiar with.”__
>
> __"Most of the paper needs to be spent on explaining your contribution, not on establishing preliminaries, as is currently the case."__
>
> To maximize accessibility we tailored our work towards providing intuition to those who may not be so comfortable with these models, which has been appreciated by other reviewers (e.g. Reviewer 2ejB, “The paper is very well written and provides a comprehensive overview of the field and the problem setup.”)
>
> __"No supplementary material"__
>
> There are 9 pages of supplementary material in the appendix.

---

### Official Review · Reviewer_ZT7H · 2025-03-17

**Overall Recommendation:** 4

**Summary:**

Estimating probability densities is a recurring task in machine learning and statistics. However, this becomes challenging in high-dimensional settings due to the curse of dimensionality, where the number of required samples grows exponentially with the dimension.

This paper addresses the challenge of high-dimensional density estimation by leveraging structured dependencies in real-world datasets such as images, videos, and text. These datasets often exhibit underlying structures that can be represented as graphs with small cliques, enabling more efficient estimation. The authors show theoretically the existence of deep neural networks trained to learn the density from samples that can achieve convergence rates that are independent of the dimension. In particular, when data dependencies are modeled by a power graph Markov random field, the neural network estimator converges at a rate of $n^{-1/(4+r)}$ in the $L^2$ norm, where $n$ is the number of samples and $r$ is the size of the largest clique in the graph of MRF. The notion of power graphs extends classical Markov random fields by capturing higher-order interactions.

## Update after rebuttal
I would like to thank the authors for their response. I am maintaining my score.

**Claims And Evidence:**

Overall, the paper is well-written, and its claims are clearly stated, well-supported, and their consequences are well discussed. A mild criticism is that while the theoretical results establish improved convergence rates when the probability distribution follows a known clique structure, they do not address how to infer this structure from data.

**Essential References Not Discussed:**

-

**Experimental Designs Or Analyses:**

N/A

**Methods And Evaluation Criteria:**

The contribution of the article is theoretical and does not rely on empirical evaluation or benchmark datasets. While this provides valuable theoretical insights, the lack of an explicit method for learning the clique structure limits its immediate practical applicability. A discussion on how these theoretical findings could be integrated into practical density estimation frameworks would strengthen the paper.

**Other Comments Or Suggestions:**

1) The paragraph beginning with 'While the manifold hypothesis...' is quite confusing. The discussion seems to conflate the concept of a data distribution being supported on a manifold with the statistical independence of samples. For instance, two independent samples from a uniform distribution on a manifold are still independent, even though they lie within the manifold. I believe it would be helpful to clarify this distinction and refine the formulation of this idea.

2) Figure 3: The visualization would be more informative if the entire dataset were used instead of just a subset.

**Other Strengths And Weaknesses:**

-

**Questions For Authors:**

1) The theoretical results assume that the clique structure of the probability distribution is given. Could you elaborate on possible approaches for inferring this structure from data?

2) Is it possible to achieve the same rate of convergence with shallow neural networks?

3) Do you think it is possible to design a kernel specifically adapted to the clique structure of the probability distribution that achieves a similar convergence rate? If so, how would such a kernel compare to the neural network approach in terms of efficiency and theoretical guarantees?

**Relation To Broader Scientific Literature:**

Density estimation is a recurring task in machine learning and statistics. This work contributes to the field by establishing dimension-independent convergence rates for structured probability densities.

**Theoretical Claims:**

I checked the proofs and did not find any glaring mistakes.

---

> ### Author Rebuttal · Authors · 2025-03-31
>
> __"A mild criticism is that while the theoretical results establish improved convergence rates when the probability distribution follows a known clique structure, they do not address how to infer this structure from data."__
>
> __"The theoretical results assume that the clique structure of the probability distribution is given...?"__
>
> A significant advantage of the settings we consider is that they naturally suggest reasonable MRF structures that can be verified empirically, as we have shown. Indeed, datatypes for which DNNs have historically shown exceptional success—such as images, audio, and text—naturally exhibit conditional dependence structures closely related to the MRF framework we propose. This connection is well-established, as evidenced by comprehensive textbooks dedicated to applying Markov Random Fields to image processing tasks [1,2] .
>
> Outside of these settings, we acknowledge that identifying the graph or $r$ from data can be challenging. With the extra space allotted for the camera ready we will include related work on the estimation of MRF graphs.
>
> __"The paragraph beginning with 'While the manifold hypothesis...' is quite confusing. The discussion seems to conflate the concept of a data distribution being supported on a manifold with the statistical independence of samples. For instance, two independent samples from a uniform distribution on a manifold are still independent, even though they lie within the manifold. I believe it would be helpful to clarify this distinction and refine the formulation of this idea."__
>
> We appreciate the reviewer’s observation and agree that the example is correct. In retrospect, our point was too vague, and we will update the paper to state this more precisely. First, we consider the manifold hypothesis:
>
> __Manifold hypothesis:__ A probability measure $\mu$ on $\mathbb{R}^d$ satisfies the “manifold hypothesis” if $\operatorname{supp}(\mu) \subset M$, where $M$ is a submanifold of $\mathbb{R}^d$ with dimension less than $d$.
>
> Here we have the following lemma makes our point relating the manifold hypothesis and dependence precisely (which is maybe of interest in its own):
>
> __Lemma:__
> Let $ M \subset \mathbb{R}^d$ be a submanifold of dimension $d' < d$. Let $X = (X_1, \ldots, X_d)$ be a random variable defined by a probability density function $p$ on $M$ (using the standard manifold area/volume measure), and assume that for each $i$, the marginal distribution of $X_i$ is given by a probability density function $p_{X_i}$ (i.e. they are nondegenerate). Then for any subset of $d'+1$ coordinates $I_1, \ldots, I_{d'+1}$, the collection of random variables $\bigl( X_{I_1}, \ldots, X_{I_{d'+1}} \bigr)$
> cannot be mutually independent.
>
> So, for any density on a $d'$-dimensional manifold embedded in Euclidean space, where the marginal distribution for each covariate is non-degenerate, any collection of $d'+1$ covariates must exhibit some dependencies. In other words, the lower the intrinsic dimension $d'$, the more dependence must be present among the covariates. The proof this is simple somewhat technical: basically if we assume $d'+1$ covariates are independent then their support $\prod_{i=1}^{d'+1}$ looks like a $d'+1$ dimensional manifold, which contradicts the $d'$ dimensional manifold assumption (we will put this into the appendix).
>
> __“Figure 3: The visualization would be more informative if the entire dataset were used instead of just a subset."__
>
> The full dataset is 60,000 points, which would be difficult to visualize, so we felt that 100 randomly selected points made this plot easier to understand. We will happily add more points if you like.
>
> __"Is it possible to achieve the same rate of convergence with shallow neural networks?"__
>
> __"“Do you think it is possible to design a kernel specifically adapted to the clique structure of the probability distribution that achieves a similar convergence rate?"__
>
> These are interesting questions! While we do not know of a shallow method or an adaptive kernel that achieves this directly, these are fascinating problems for future work. We hope that our results provide a foundation for further investigation into how and why dimension-independent rates are attainable in models for images, audio, and text.
>
>
> [1] Li, S. Z. Markov Random Field Modeling in Image Analysis, 2009 (3000+ citations)
>
> [2] Blake, A., Kohli, P., and Rother, C. Markov random fields for vision and image processing. 2011

---

> > ### Comment · Reviewer_ZT7H · 2025-04-02
> >
> > Thank you for your rebuttal. Please update the manuscript accordingly.

---

> > > ### Author Response · Authors · 2025-04-03
> > >
> > > Thank you for the acknowledgement! We are happy to update the manuscript. Although ICML does not allow revisions at this stage, we will certainly include these revisions in the camera ready version.

---

### Decision · Program_Chairs · 2025-05-01

**Decision:**

Accept (poster)

**Comment:**

The reviews are mostly positive, and the only negative review has little concrete criticism. Using Neural Networks to learn MRFs is a novel idea, and the paper proves that DNNs can leverage this structure efficiently. I am in favor of accepting this paper.